# Changes in the associations of race and rurality with SARS-CoV-2 infection, mortality, and case fatality in the United States from February 2020 to March 2021: A population-based cohort study

**George N. Ioannou**[1,2]*, **Jacqueline M. Ferguson**[3], **Ann M. O'Hare**[4], **Amy S. B. Bohnert**[5], **Lisa I. Backus**[6], **Edward J. Boyko**[7], **Thomas F. Osborne**[8], **Matthew L. Maciejewski**[9,10,11,12], **C. Barrett Bowling**[13], **Denise M. Hynes**[14,15], **Theodore J. Iwashyna**[16,17], **Melody Saysana**[2], **Pamela Green**[2], **Kristin Berry**[2]

1 Divisions of Gastroenterology, Veterans Affairs Puget Sound Healthcare System and University of Washington, Seattle, Washington, United States of America, 2 Research and Development, Veterans Affairs Puget Sound Health Care System, Seattle, Washington, United States of America, 3 Center for Innovation to Implementation, VA Palo Alto Healthcare System, US Department of Veterans Affairs, Palo Alto, California, United States of America, 4 Nephrology, Veterans Affairs Puget Sound Healthcare System and University of Washington, Seattle, Washington, United States of America, 5 Department of Psychiatry, University of Michigan Medical School, Ann Arbor, Michigan, United States of America, 6 Department of Veterans Affairs, Population Health, Palo Alto Healthcare System, Palo Alto, California, United States of America, 7 General Internal Medicine, Veterans Affairs Puget Sound Healthcare System and University of Washington, Seattle, Washington, United States of America, 8 Veterans Affairs Palo Alto Healthcare System, Palo Alto, and Department of Radiology, Stanford University School of Medicine, Stanford, California, United States of America, 9 Center of Innovation to Accelerate Discovery and Practice Transformation (ADAPT), Durham Veterans Affairs Health Care System, Durham, North Carolina, United States of America, 10 Department of Population Health Sciences, Duke University School of Medicine, Durham, North Carolina, United States of America, 11 Duke-Margolis Center for Health Policy, Duke University School of Medicine, Durham, North Carolina, United States of America, 12 Division of General Internal Medicine, Duke University School of Medicine, Durham, North Carolina, United States of America, 13 Durham Veterans Affairs Geriatric Research Education and Clinical Center, Durham Veterans Affairs Medical Center (VAMC), Durham, NC and Department of Medicine, Duke University, Durham, North Carolina, United States of America, 14 Center of Innovation to Improve Veteran Involvement in Care, VA Portland Healthcare System, Portland, Oregon, United States of America, 15 Health Management and Policy, School of Social and Behavioral Health Sciences, College of Public Health and Human Sciences, Health Data and Informatics Program, Center for Genome Research and Biocomputing, Oregon State University, Corvallis, Oregon, United States of America, 16 Center for Clinical Management Research, VA Ann Arbor Health System, Ann Arbor, Michigan, United States of America, 17 Division of Pulmonary and Critical Care Medicine, Department of Internal Medicine, University of Michigan, Ann Arbor, Michigan, United States of America

* georgei@medicine.washington.edu

**Data Availability Statement:** Data cannot be shared publicly because the Department of

## Abstract

### Background

We examined whether key sociodemographic and clinical risk factors for Severe Acute Respiratory Syndrome Coronavirus 2 (SARS-CoV-2) infection and mortality changed over time in a population-based cohort study.

### Methods and findings

In a cohort of 9,127,673 persons enrolled in the United States Veterans Affairs (VA) healthcare system, we evaluated the independent associations of sociodemographic and clinical

Veterans Affairs prevents public sharing of national VA EHR data. Data are available only to VA investigators who obtain the required IRB approvals to access the data. For access to data please look at the current contact information for the Seattle-Denver Veterans Affairs Health Services Research and Development (HSR&D) Center of Innovation (COIN) at: https://www.hsrd.research.va.gov/centers/seattle-denver.cfm.

**Funding:** The study was supported by the U.S. Department of Veterans Affairs, CSR&D grant COVID19-8900-11 to GI. The study was supported by the U.S. Department of Veterans Affairs HSR&D grant C19 21-278 to GI, AB, EB and MM. The study was supported by the U.S. Department of Veterans Affairs HSR&D grant C19 21-279 to TI, DH, AO and CBB. MM was also supported by a Research Career Scientist award from the Department of Veterans Affairs (RCS 10-391) and by the Center of Innovation to Accelerate Discovery and Practice Transformation (CIN 13-410) at the Durham VA Health Care System. The funders had no role in study design, data collection and analysis, decision to publish, or preparation of the manuscript.

**Competing interests:** I have read the journal's policy and the authors of this manuscript have the following competing interests: DH reported that she is an employee of the Department of Veterans Affairs. The remaining authors have declared no competing interests exist.

**Abbreviations:** AI/AN, American Indian/Alaska Native; AOR, adjusted odds ratio; BIRLS, Beneficiary Identification and Records Locator System; BMI, body mass index; CCI, Charlson comorbidity index; CDW, Corporate Data Warehouse; COVID-19, Coronavirus Disease 2019; EHR, electronic health record; IRB, Institutional Review Board; PCR, polymerase chain reaction; PI/NH, Pacific Islander/Native Hawaiian; RUCA, Rural–Urban Commuting Area; SARS-CoV-2, Severe Acute Respiratory Syndrome Coronavirus 2; SSA, Social Security Administration; STROBE, Strengthening the Reporting of Observational Studies in Epidemiology; VA, Veterans Affairs; VINCI, VA Informatics and Computing Infrastructure.

characteristics with SARS-CoV-2 infection ($n$ = 216,046), SARS-CoV-2–related mortality ($n$ = 10,230), and case fatality at monthly intervals between February 1, 2020 and March 31, 2021. VA enrollees had a mean age of 61 years (SD 17.7) and were predominantly male (90.9%) and White (64.5%), with 14.6% of Black race and 6.3% of Hispanic ethnicity. Black (versus White) race was strongly associated with SARS-CoV-2 infection (adjusted odds ratio [AOR] 5.10, [95% CI 4.65 to 5.59], $p$-value <0.001), mortality (AOR 3.85 [95% CI 3.30 to 4.50], $p$-value < 0.001), and case fatality (AOR 2.56, 95% CI 2.23 to 2.93, $p$-value < 0.001) in February to March 2020, but these associations were attenuated and not statistically significant by November 2020 for infection (AOR 1.03 [95% CI 1.00 to 1.07] $p$-value = 0.05) and mortality (AOR 1.08 [95% CI 0.96 to 1.20], $p$-value = 0.21) and were reversed for case fatality (AOR 0.86, 95% CI 0.78 to 0.95, $p$-value = 0.005). American Indian/Alaska Native (AI/AN versus White) race was associated with higher risk of SARS-CoV-2 infection in April and May 2020; this association declined over time and reversed by March 2021 (AOR 0.66 [95% CI 0.51 to 0.85] $p$-value = 0.004). Hispanic (versus non-Hispanic) ethnicity was associated with higher risk of SARS-CoV-2 infection and mortality during almost every time period, with no evidence of attenuation over time. Urban (versus rural) residence was associated with higher risk of infection (AOR 2.02, [95% CI 1.83 to 2.22], $p$-value < 0.001), mortality (AOR 2.48 [95% CI 2.08 to 2.96], $p$-value < 0.001), and case fatality (AOR 2.24, 95% CI 1.93 to 2.60, $p$-value < 0.001) in February to April 2020, but these associations attenuated over time and reversed by September 2020 (AOR 0.85, 95% CI 0.81 to 0.89, $p$-value < 0.001 for infection, AOR 0.72, 95% CI 0.62 to 0.83, $p$-value < 0.001 for mortality and AOR 0.81, 95% CI 0.71 to 0.93, $p$-value = 0.006 for case fatality). Throughout the observation period, high comorbidity burden, younger age, and obesity were consistently associated with infection, while high comorbidity burden, older age, and male sex were consistently associated with mortality. Limitations of the study include that changes over time in the associations of some risk factors may be affected by changes in the likelihood of testing for SARS-CoV-2 according to those risk factors; also, study results apply directly to VA enrollees who are predominantly male and have comprehensive healthcare and need to be confirmed in other populations.

## Conclusions

In this study, we found that strongly positive associations of Black and AI/AN (versus White) race and urban (versus rural) residence with SARS-CoV-2 infection, mortality, and case fatality observed early in the pandemic were ameliorated or reversed by March 2021.

## Author summary

### Why was this study done?

- As the Coronavirus Disease 2019 (COVID-19) pandemic continues to evolve, some risk factors for infection with COVID-19 and death due to COVID-19 that were described early in the pandemic may be changing.

- Recognizing such changes is important in informing population-based approaches to prevent infection and reduce mortality.

## What did the researchers do and find?

- We investigated how the associations of key sociodemographic and clinical factors with COVID-19 infection, mortality, or case fatality changed between February 2020 and March 2021 among a cohort of approximately 9.1 million persons enrolled in the national US Veterans Affairs (VA) healthcare system, including 216,046 who tested positive and 10,230 who died of COVID-19 during the study period.

- Black (versus White) race was strongly associated with a 5-fold higher risk of Severe Acute Respiratory Syndrome Coronavirus 2 (SARS-CoV-2) infection, a 4-fold higher risk of mortality, and a 2.5-fold higher risk of case fatality in February to March 2020, but these associations attenuated over time and were no longer statistically significant by November 2020 for infection and mortality and were reversed for case fatality.

- American Indian/Alaska Native (AI/AN versus White) race was associated with SARS-CoV-2 infection early in the pandemic, but this association declined over time and reversed by March 2021.

- Urban (versus rural) residence was associated with 2-fold higher risk of infection, a 2.5-fold higher risk of mortality, and 2.2-fold higher risk of case fatality in February to April 2020, but these associations attenuated over time and reversed by September 2020.

- Throughout the observation period, high comorbidity burden, younger age, Hispanic ethnicity, and obesity were consistently associated with infection, while high comorbidity burden, older age, Hispanic ethnicity, and male sex were consistently associated with mortality.

## What do these findings mean?

- Early in the pandemic, there were strongly positive associations of Black and AI/AN (versus White) race and urban (versus rural) residence with SARS-CoV-2 infection, mortality, and case fatality, but these were ameliorated or even reversed by March 2021.

- Our results apply directly to VA enrollees who are predominantly male and have access to universal healthcare; they need to be confirmed in other populations.

## Introduction

Sociodemographic factors and comorbidity burden have emerged as major risk factors for Severe Acute Respiratory Syndrome Coronavirus 2 (SARS-CoV-2) infection and mortality [1–13]. Black, American Indian/Alaska Native (AI/AN), and Hispanic persons have been reported to have higher risk of SARS-CoV-2 infection and mortality than White and non-Hispanic persons [7–11,14]. Obese persons [12,15] and those with higher comorbidity burden [9] have also

been reported to have higher risk of SARS-CoV-2 infection and mortality, while older age is one of the strongest risk factors for SARS-CoV-2–related mortality [2,9]. Over the course of the pandemic, residing in a geographical region with high incidence of SARS-CoV-2 infection at a given time has proven to be a strong risk factor for SARS-CoV-2 infection and mortality [2,5,16].

Three "waves" of the SARS-CoV-2 pandemic have been described in the US, with peaks in cases in April 2020 (first wave), July 2020 (second wave), and December 2020 to January 2021 (third wave). Risk factors for SARS-CoV-2–related infection and mortality may be changing over time since the pandemic began, especially in relation to these waves. Although it is clear that geographical regions with high infection, hospitalization, and mortality rates changed over time as the pandemic surged in different parts of the country, prior studies have not examined whether risk factors such as age, comorbidity burden, race, and ethnicity also varied. Over the course of the pandemic, there have been marked changes in use of prophylactic measures and access to care as well as viral characteristics (e.g., emergence of new variants) and availability of treatments (e.g., use of different pharmacotherapies). These changes may have affected the associations of sociodemographic and other risk factors with SARS-CoV-2 infection or mortality. Understanding changing patterns of risk factors could be important in informing population-based approaches to prevent infection and reduce mortality by targeting those at highest risk at any given time during the time course of an evolving pandemic. Also, changing patterns over time in sociodemographic risk factors for SARS-CoV-2 may provide insights that are more broadly applicable to disparities research in general.

We identified a cohort of approximately 9.1 million persons who were enrolled in the national US Department of Veterans Affairs (VA) healthcare system at the beginning of the pandemic in February 2020 and followed this cohort over the ensuing 14-month period to determine whether the magnitude and direction of the associations of established risk factors with SARS-CoV-2 infection, mortality, and case fatality changed over time.

## Methods

### Study population and data source

The VA supports the largest integrated national healthcare system in the US, providing care at 168 medical centers and 1,112 outpatient clinics throughout the country. We identified a cohort of all persons aged $\geq$18 years who were alive and enrolled in the VA healthcare system on February 1, 2020 ($n$ = 9,127,673). We followed this cohort for the development of SARS-CoV-2 infection and SARS-CoV-2–related mortality until March 31, 2021.

The VA employs a comprehensive, nationwide electronic health record (EHR) system. EHR data from all VA facilities are transferred to the VA's centralized relational database, the Corporate Data Warehouse (CDW), on a nightly basis to support research and clinical operations [17]. The CDW includes the "COVID-19 Shared Data Resource," a set of analytic variables and datasets related to Coronavirus Disease 2019 (COVID-19) developed and maintained by the VA Informatics and Computing Infrastructure (VINCI) specifically to facilitate COVID-19 research and operations, which we used in combination with other CDW data.

The study was approved by the VA Puget Sound Institutional Review Board (IRB# 01885), which waived the requirement to obtain informed consent because this was a retrospective study of data from an existing database.

### Definition of SARS-CoV-2 infection and SARS-CoV-2–related death

Cohort members who tested positive for SARS-CoV-2 within the VA system based on polymerase chain reaction (PCR) tests were defined as having infection. The earliest date of a

documented positive test was taken as each patient's date of infection. Patients who died of any cause within 30 days of infection were defined as having a SARS-CoV-2–related death consistent with prior studies [2,3]. Deaths occurring both within and outside the VA are comprehensively captured in CDW from a variety of VA and non-VA sources including VA inpatient files, VA Beneficiary Identification and Records Locator System (BIRLS), Social Security Administration (SSA) death files, and the Department of Defense [18]. Deaths occurring from February 1, 2020 to February 28, 2021 were ascertained from files that were updated on June 25, 2021 to allow time for deaths to be electronically recorded in CDW.

## Study outcomes: Monthly SARS-CoV-2 infection rates, mortality rates, and case fatality rates

SARS-CoV-2 infection and mortality rates were calculated monthly as a proportion of all cohort members who were still alive and at risk on the first day of each month. For the monthly analysis of SARS-CoV-2 infection rates, we excluded persons who died of any cause or were infected with SARS-CoV-2 before the beginning of each monthly period. For the analysis of SARS-CoV-2 mortality rates, we excluded persons who died of any cause before the beginning of each monthly period or were infected with SARS-CoV-2 more than 30 days before the beginning of the month (since SARS-CoV-2–related mortality was defined as death within 30 days of infection). The aim of removing from our cohort persons who were no longer at risk at the beginning of each observation period was to avoid a spurious attenuation of risk factors over time due to "depletion" of at-risk persons.

Monthly case fatality rates were calculated as the proportion of the patients who tested positive each month who died of any cause within 30 days of the earliest positive test.

## Sociodemographic factors and comorbidity burden

We ascertained the following 8 characteristics for all cohort members on the first day of each monthly observation period: age (categorized as shown in **Table 1** or modeled as restricted cubic splines with 5 knots at ages 30, 49, 64, 73, and 88 years corresponding to 5, 27.5, 50, 72.5, and 95 percentiles as recommended [19]), sex, self-reported race (White, Black, Asian, AI/AN, Pacific Islander/Native Hawaiian [PI/NH], and Other) and ethnicity (Hispanic and non-Hispanic), urban versus rural residence (based on zip codes, using data from the VA Office of Rural Health [20], which uses the Secondary Rural–Urban Commuting Area [RUCA] for defining rurality), geographic location (divided into 10 standard US Federal Regions [21]), body mass index (BMI, categorized using the World Health Organization groups [22] as shown in **Table 1** or modeled as restricted cubic splines with 5 knots at BMIs of 21.3, 26.0, 29.0, 32.5, and 39.9 kg/m$^2$ corresponding to 5, 27.5, 50, 72.5, and 95 percentiles), and Charlson comorbidity index (CCI) [23] calculated using the Deyo modification [24], which takes into account 19 comorbid conditions reported in the 2 years prior to each observation period. We focused on these sociodemographic factors, obesity, and the CCI because they are some of the most important risk factors for SARS-CoV-2 infection or mortality reported in the literature [1–13] and because trends in the associations over time could be plausibly hypothesized.

Missing values for BMI (20.9%) were multiply imputed using values of the other covariates included in multivariable analyses. "Missing" values for race (18.7%) or ethnicity (15.6%) included persons who refused to declare their race/ethnicity or reported unknown or mixed race/ethnicity and did not self-identify as belonging to one of the prespecified racial or ethnic group. For these persons, we did not perform imputation of race/ethnicity, but rather included them in a "missing/unknown/refused" category.

**Table 1. Cohort characteristics and incidence of SARS-CoV-2 infection presented by month in a cohort of 9.1 million VA enrollees followed from February 2020 to March 2021.**

| Time period and number of VA enrollees at risk* | Cohort characteristics N (%) | Number of SARS-CoV-2–positive persons (and incidence per 10,000 persons per month) | | | | | | | | | | | | |
|---|---|---|---|---|---|---|---|---|---|---|---|---|---|---|
| | Entire period: February 2020 to March 2021 N = 9,127,673 | February to March 2020 N = 9,090,196 | April 2020 N = 9,053,082 | May 2020 N = 9,022,051 | June 2020 N = 8,990,833 | July 2020 N = 8,948,674 | August 2020 N = 8,913,864 | September 2020 N = 8,881,573 | October 2020 N = 8,844,680 | November 2020 N = 8,804,424 | December 2020 N = 8,744,388 | January 2021 N = 8,671,916 | February 2021 N = 8,630,283 | March 2021 |
| **All persons infected** | 216,046 (100%) | 2,470 (2.7) | 7,346 (8.1) | 4,957 (5.5) | 7,634 (8.5) | 16,105 (17.9) | 9,189 (10.3) | 7,526 (8.4) | 13,523 (15.2) | 33,553 (37.9) | 47,357 (53.8) | 39,431 (45.1) | 17,698 (20.4) | 9,257 (10.7) |
| **Sex** | | | | | | | | | | | | | | |
| Female | 829,355 (9.1) | 218 (2.6) | 695 (8.4) | 453 (5.5) | 853 (10.3) | 1,946 (23.6) | 968 (11.8) | 716 (8.7) | 1,243 (15.2) | 3,315 (40.6) | 4,859 (59.7) | 3,942 (48.7) | 1,825 (22.7) | 1,003 (12.5) |
| Male | 8,298,318 (90.9) | 2,252 (2.7) | 6,651 (8.1) | 4,504 (5.5) | 6,781 (8.3) | 14,159 (17.3) | 8,221 (10.1) | 6,810 (8.4) | 12,280 (15.2) | 30,238 (37.7) | 42,498 (53.2) | 35,489 (44.7) | 15,873 (20.2) | 8,254 (10.5) |
| **Age (years)** | | | | | | | | | | | | | | |
| 18 to 24 | 81,085 (0.9) | 8 (1.0) | 26 (3.2) | 16 (2.0) | 88 (10.9) | 153 (18.9) | 66 (8.2) | 42 (5.2) | 69 (8.6) | 207 (25.7) | 303 (37.7) | 229 (28.6) | 86 (10.8) | 59 (7.4) |
| 25 to 34 | 828,607 (9.1) | 144 (1.7) | 388 (4.7) | 273 (3.3) | 919 (11.1) | 1,681 (20.3) | 737 (8.9) | 538 (6.5) | 993 (12.1) | 2,648 (32.2) | 3,633 (44.3) | 2,745 (33.6) | 1,220 (15.0) | 767 (9.4) |
| 35 to 44 | 1,069,496 (11.7) | 279 (2.6) | 614 (5.7) | 405 (3.8) | 1,005 (9.4) | 2,168 (20.3) | 1,018 (9.6) | 794 (7.5) | 1,488 (14.0) | 3,898 (36.7) | 5,409 (51.2) | 4,382 (41.7) | 1,879 (17.9) | 1,205 (11.5) |
| 45 to 54 | 1,175,283 (12.9) | 369 (3.1) | 860 (7.3) | 546 (4.7) | 1,174 (10.0) | 2,708 (23.1) | 1,364 (11.7) | 1,103 (9.5) | 1,952 (16.8) | 4,938 (42.5) | 6,790 (58.7) | 5,524 (48.1) | 2,455 (21.5) | 1,496 (13.1) |
| 55 to 64 | 1,517,143 (16.6) | 562 (3.7) | 1,514 (10.0) | 1,061 (7.0) | 1,420 (9.4) | 3,086 (20.5) | 1,810 (12.1) | 1,385 (9.3) | 2,506 (16.8) | 6,093 (40.9) | 8,846 (59.7) | 7,579 (51.5) | 3,478 (23.8) | 1,927 (13.2) |
| 65 to 74 | 2,434,839 (26.7) | 688 (2.8) | 2,111 (8.7) | 1,399 (5.8) | 1,808 (7.5) | 3,989 (16.6) | 2,554 (10.7) | 2,215 (9.3) | 3,938 (16.6) | 9,484 (40.1) | 13,197 (56.1) | 11,319 (48.5) | 5,167 (22.3) | 2,355 (10.2) |
| 75 to 84 | 1,257,097 (13.8) | 279 (2.2) | 1,022 (8.2) | 693 (5.6) | 795 (6.5) | 1,564 (12.8) | 1,123 (9.2) | 999 (8.3) | 1,797 (15.0) | 4,389 (36.8) | 6,368 (53.7) | 5,380 (45.7) | 2,384 (20.5) | 1,050 (9.1) |
| ≥85 | 764,123 (8.4) | 141 (1.8) | 811 (10.8) | 564 (7.7) | 425 (5.9) | 756 (10.6) | 517 (7.3) | 450 (6.5) | 780 (11.4) | 1,896 (28.0) | 2,811 (41.9) | 2,273 (34.2) | 1,029 (15.9) | 398 (6.2) |
| **Race** | | | | | | | | | | | | | | |
| White | 5,886,250 (64.5) | 956 (1.6) | 3,774 (6.4) | 2,739 (4.7) | 4,414 (7.6) | 9,315 (16.1) | 5,690 (9.9) | 5,141 (9.0) | 9,898 (17.3) | 24,753 (43.5) | 33,126 (58.5) | 26,626 (47.4) | 11,932 (21.4) | 6,425 (11.6) |
| Black | 1,337,163 (14.6) | 1,293 (9.7) | 2,919 (21.9) | 1,729 (13.0) | 2,319 (17.6) | 4,922 (37.4) | 2,544 (19.4) | 1,602 (12.3) | 2,267 (17.4) | 5,337 (41.2) | 8,959 (69.5) | 8,488 (66.4) | 3,909 (30.8) | 1,914 (15.2) |
| Asian | 114,794 (1.3) | 27 (2.4) | 55 (4.8) | 37 (3.2) | 75 (6.6) | 168 (14.7) | 81 (7.1) | 70 (6.2) | 95 (8.4) | 278 (24.6) | 550 (48.8) | 442 (39.4) | 191 (17.1) | 82 (7.4) |
| AI/AN | 79,738 (0.9) | 14 (1.8) | 54 (6.8) | 48 (6.1) | 92 (11.6) | 155 (19.7) | 79 (10.1) | 76 (9.7) | 147 (18.9) | 340 (43.9) | 439 (56.9) | 349 (45.6) | 162 (21.3) | 60 (7.9) |
| PI/NH | 75,186 (0.8) | 13 (1.7) | 57 (7.6) | 44 (5.9) | 77 (10.3) | 157 (21.1) | 101 (13.7) | 73 (9.9) | 115 (15.7) | 301 (41.1) | 440 (60.4) | 393 (54.3) | 177 (24.7) | 68 (9.5) |
| Missing/ unknown/ refused | 1,634,542 (17.9) | 167 (1.0) | 487 (3.0) | 360 (2.2) | 657 (4.1) | 1,388 (8.6) | 694 (4.3) | 564 (3.5) | 1,001 (6.3) | 2,544 (16.0) | 3,843 (24.2) | 3,133 (19.8) | 1,327 (8.4) | 708 (4.5) |
| **Ethnicity** | | | | | | | | | | | | | | |
| Non-Hispanic | 7,201,109 (78.9) | 2,119 (2.9) | 6,441 (9.0) | 4,365 (6.1) | 6,101 (8.6) | 12,936 (18.3) | 7,825 (11.1) | 6,615 (9.4) | 11,868 (17.0) | 29,477 (42.3) | 40,749 (58.8) | 33,980 (49.4) | 15,482 (22.7) | 8,014 (11.8) |
| Hispanic | 571,236 (6.3) | 272 (4.8) | 649 (11.4) | 426 (7.5) | 1,235 (21.8) | 2,583 (45.7) | 1,037 (18.5) | 645 (11.5) | 1,159 (20.8) | 2,819 (50.7) | 4,648 (84.1) | 3,791 (69.2) | 1,487 (27.4) | 861 (15.9) |
| Missing/ unknown/ refused | 1,355,328 (14.8) | 79 (0.6) | 256 (1.9) | 166 (1.2) | 298 (2.2) | 586 (4.4) | 327 (2.4) | 266 (2.0) | 496 (3.7) | 1,257 (9.5) | 1,960 (14.8) | 1,660 (12.6) | 729 (5.6) | 382 (2.9) |
| **US Federal Region†** | | | | | | | | | | | | | | |
| 1 | 400,339 (4.4) | 95 (2.4) | 852 (21.4) | 398 (10.1) | 196 (5.0) | 143 (3.6) | 102 (2.6) | 99 (2.5) | 219 (5.6) | 946 (24.5) | 1,674 (43.4) | 1,520 (39.7) | 760 (20.0) | 419 (11.1) |
| 2 | 665,463 (7.3) | 615 (9.2) | 1,717 (26.0) | 590 (9.0) | 267 (4.1) | 340 (5.2) | 217 (3.4) | 209 (3.2) | 380 (5.9) | 1,248 (19.5) | 2,183 (34.2) | 2,114 (33.3) | 1,079 (17.1) | 759 (12.1) |
| 3 | 958,643 (10.5) | 185 (1.9) | 838 (8.8) | 657 (6.9) | 459 (4.8) | 773 (8.2) | 581 (6.2) | 474 (5.1) | 830 (8.9) | 2,528 (27.1) | 4,474 (48.2) | 3,725 (40.4) | 1,815 (19.8) | 1,048 (11.5) |
| 4 | 2,290,207 (25.1) | 363 (1.6) | 1,085 (4.8) | 1,006 (4.4) | 2,474 (10.9) | 5,993 (26.5) | 3,416 (15.2) | 2,328 (10.4) | 3,045 (13.7) | 5,555 (25.0) | 11,123 (50.3) | 11,489 (52.3) | 5,354 (24.6) | 2,470 (11.4) |
| 5 | 1,325,949 (14.5) | 413 (3.1) | 1,230 (9.3) | 955 (7.3) | 605 (4.6) | 1,217 (9.3) | 1,007 (7.8) | 1,095 (8.5) | 3,058 (23.7) | 8,438 (65.8) | 7,528 (59.2) | 4,576 (36.3) | 2,187 (17.5) | 1,436 (11.5) |
| 6 | 1,194,444 (13.1) | 459 (3.8) | 658 (5.5) | 486 (4.1) | 1,744 (14.7) | 4,019 (34.1) | 1,687 (14.4) | 1,206 (10.3) | 2,161 (18.6) | 4,403 (38.0) | 6,419 (55.7) | 6,182 (54.0) | 2,338 (20.6) | 1,104 (9.8) |
| 7 | 458,363 (5.0) | 58 (1.3) | 238 (5.2) | 250 (5.5) | 221 (4.9) | 600 (13.3) | 681 (15.2) | 853 (19.1) | 1,432 (32.2) | 3,652 (82.6) | 3,057 (69.8) | 2,007 (46.2) | 935 (21.7) | 465 (10.9) |
| 8 | 374,484 (4.1) | 74 (2.0) | 208 (5.6) | 179 (4.8) | 147 (4.0) | 277 (7.5) | 241 (6.5) | 414 (11.3) | 1,100 (30.1) | 2,568 (70.6) | 1,908 (52.8) | 992 (27.7) | 515 (14.4) | 369 (10.4) |
| 9 | 1,041,079 (11.4) | 164 (1.6) | 387 (3.7) | 344 (3.3) | 1,376 (13.3) | 2,356 (22.9) | 967 (9.5) | 627 (6.2) | 860 (8.5) | 3,112 (30.7) | 7,725 (76.6) | 6,059 (60.6) | 2,253 (22.8) | 875 (8.9) |
| 10 | 418,702 (4.6) | 44 (1.1) | 133 (3.2) | 92 (2.2) | 145 (3.5) | 387 (9.3) | 290 (7.0) | 221 (5.4) | 438 (10.7) | 1,103 (27.0) | 1,266 (31.1) | 767 (18.9) | 462 (11.4) | 312 (7.8) |

*(Continued)*

**Table 1.** (Continued)

| Time period and number of VA enrollees at risk[*] | Cohort characteristics N (%) Entire period: February 2020 to March 2021 N = 9,127,673 | Number of SARS-CoV-2–positive persons (and incidence per 10,000 persons per month) | | | | | | | | | | | | |
|---|---|---|---|---|---|---|---|---|---|---|---|---|---|---|
| | | February to March 2020 N = 9,127,673 | April 2020 N = 9,090,196 | May 2020 N = 9,053,082 | June 2020 N = 9,022,051 | July 2020 N = 8,990,833 | August 2020 N = 8,948,674 | September 2020 N = 8,913,864 | October 2020 N = 8,881,573 | November 2020 N = 8,844,680 | December 2020 N = 8,804,424 | January 2021 N = 8,744,388 | February 2021 N = 8,671,916 | March 2021 N = 8,630,283 |
| **Urban versus rural** | | | | | | | | | | | | | | |
| Rural | 4,469,258 (49.0) | 579 (1.3) | 1,870 (4.2) | 1,623 (3.7) | 2,955 (6.7) | 6,872 (15.6) | 4,680 (10.7) | 4,147 (9.5) | 7,468 (17.2) | 17,506 (40.4) | 23,772 (55.1) | 19,440 (45.4) | 8,734 (20.6) | 4,513 (10.7) |
| Urban | 4,658,415 (51.0) | 1,891 (4.1) | 5,476 (11.8) | 3,334 (7.2) | 4,679 (10.2) | 9,233 (20.1) | 4,509 (9.9) | 3,379 (7.4) | 6,055 (13.4) | 16,047 (35.6) | 23,585 (52.5) | 19,991 (44.8) | 8,964 (20.3) | 4,744 (10.8) |
| **BMI (kg/m$^2$)** | | | | | | | | | | | | | | |
| <18.5 (underweight) | 64,634 (0.7) | 38 (5.9) | 148 (23.5) | 101 (16.4) | 95 (15.7) | 115 (19.3) | 98 (16.7) | 74 (12.8) | 105 (18.4) | 223 (39.6) | 374 (66.9) | 367 (66.4) | 159 (29.4) | 89 (16.6) |
| 18.5 to <25 (normal weight) | 1,841,422 (20.2) | 413 (2.2) | 1,448 (7.9) | 1,030 (5.7) | 1,218 (6.7) | 2,414 (13.4) | 1,362 (7.6) | 1,171 (6.6) | 1,864 (10.5) | 4,658 (26.5) | 6,929 (39.5) | 5,908 (33.9) | 2,703 (15.7) | 1,345 (7.8) |
| 25 to <30 (overweight) | 3,314,997 (36.3) | 794 (2.4) | 2,348 (7.1) | 1,602 (4.9) | 2,509 (7.7) | 5,183 (15.9) | 2,920 (9.0) | 2,460 (7.6) | 4,410 (13.6) | 10,898 (33.9) | 15,526 (48.4) | 12,829 (40.3) | 5,759 (18.2) | 2,928 (9.3) |
| 30 to <35 (obese I) | 2,433,428 (26.7) | 677 (2.8) | 1,901 (7.8) | 1,260 (5.2) | 2,181 (9.0) | 4,684 (19.5) | 2,679 (11.2) | 2,163 (9.1) | 3,969 (16.7) | 9,903 (41.7) | 13,689 (57.9) | 11,498 (49.0) | 5,096 (21.9) | 2,744 (11.8) |
| 35 to <40 (obese II) | 1,026,570 (11.2) | 355 (3.5) | 946 (9.2) | 633 (6.2) | 1,039 (10.2) | 2,358 (23.2) | 1,398 (13.8) | 1,031 (10.2) | 2,007 (20.0) | 4,995 (49.9) | 6,941 (69.7) | 5,696 (57.6) | 2,540 (25.9) | 1,371 (14.1) |
| ≥40 (obese III) | 446,622 (4.9) | 193 (4.3) | 555 (12.5) | 331 (7.5) | 592 (13.4) | 1,351 (30.6) | 732 (16.7) | 627 (14.3) | 1,168 (26.8) | 2,876 (66.3) | 3,898 (90.6) | 3,133 (73.5) | 1,441 (34.2) | 780 (18.6) |
| **CCI** | | | | | | | | | | | | | | |
| 0 to 1 | 6,063,266 (66.4) | 955 (1.6) | 2,620 (4.3) | 1,805 (3.0) | 3,799 (6.3) | 7,918 (13.2) | 4,197 (7.0) | 3,313 (5.5) | 6,130 (10.3) | 15,701 (26.4) | 21,571 (36.3) | 17,532 (29.7) | 8,062 (13.7) | 4,708 (8.0) |
| 2 to 3 | 1,520,767 (16.7) | 552 (3.6) | 1,649 (10.9) | 1,154 (7.7) | 1,575 (10.5) | 3,703 (24.8) | 2,093 (14.1) | 1,813 (12.3) | 3,253 (22.2) | 7,929 (54.3) | 11,334 (78.1) | 9,722 (67.7) | 4,332 (30.5) | 2,198 (15.6) |
| 4 to 5 | 768,966 (8.4) | 346 (4.5) | 1,153 (15.1) | 807 (10.7) | 985 (13.1) | 1,990 (26.6) | 1,301 (17.5) | 1,073 (14.6) | 1,815 (24.8) | 4,613 (63.4) | 6,557 (90.9) | 5,671 (79.5) | 2,427 (34.6) | 1,110 (15.9) |
| ≥6 | 774,674 (8.5) | 617 (8.0) | 1,924 (25.2) | 1,191 (15.8) | 1,275 (17.1) | 2,494 (33.8) | 1,598 (21.9) | 1,327 (18.4) | 2,325 (32.7) | 5,310 (75.5) | 7,895 (113.4) | 6,506 (94.9) | 2,877 (42.9) | 1,241 (18.7) |

[*] VA enrollees at risk are those who are still alive and not yet infected at the beginning of each time period.

[†] Categorized according to the 10 US Federal Regions drawn up by the Office of Management and Budget: 1 (CT, MA, ME, NH, RI, and VT), 2 (NJ, NY, PR, and Virgin Island), 3 (DC, DE, MD, PA, VA, and WV), 4 (AL, FL, GA, KY, MS, NC, SC, and TN), 5 (IL, IN, MI, MN, OH, and WI), 6 (AR, LA, NM, OK, and TX), 7 (IA, KS, MO, and NE), 8 (CO, MT, ND, SD, UT, and WY), 9 (AZ, CA, GU, HI, and NV), and 10 (AK, ID, OR, and WA).

AI/AN, American Indian/Alaska Native; CCI, Charlson comorbidity index; PI/NH, Pacific Islander/Native Hawaiian; SARS-CoV-2, Severe Acute Respiratory Syndrome Coronavirus 2; VA, Veterans Affairs.

## Statistical analysis

We calculated the number of new SARS-CoV-2 infections and deaths each month and the monthly incidence as a proportion of all persons in our cohort who were still alive and at risk on the first day of the month. Monthly case fatality was calculated as the proportion of patients who tested positive each month who died within 30 days. For each monthly observation period (for infections) or bimonthly observation period (for mortality and case fatality), we used a separate multivariable logistic regression model to simultaneously adjust for the 8 characteristics listed above to determine the associations between each risk factor and SARS-CoV-2 infection, mortality, or case fatality reported as an adjusted odds ratio (AOR). We used bimonthly periods for trends in mortality and case fatality because there were too few deaths in some subgroups to generate reasonably precise monthly analyses of time trends.

We used a Wald test with cluster-robust standard errors to formally evaluate whether the associations between risk factors and each outcome (infection, mortality, or case fatality) changed over time by creating another model that combined all time periods and included an interaction term separately for each risk factor (risk factor * time period) where time period was an ordinal variable (see **S1** and **S2** Tables).

Almost all analyses were proposed a priori as shown in the S1 Analytic Plan. Notable exceptions were the analyses of case fatality as an outcome and additionally modeling age and BMI as restricted cubic splines that were performed in response to reviewers' comments.

We followed the Strengthening the Reporting of Observational Studies in Epidemiology (STROBE) reporting guidelines as summarized in S1 STROBE Checklist.

## Results

### Baseline characteristics of VA enrollees

The baseline characteristics of the cohort of VA enrollees ($n$ = 9,127,673) in February 2020 are shown in **Table 1**. Mean and median age were 61.0 and 64 years, respectively, with a substantial proportion in categories 65 to 74 (26.7%), 75 to 84 (13.8%), and ≥85 (8.4%) years of age. Most cohort members were male (90.9%) and White (64.5%), with 14.6% of Black race and 6.3% of Hispanic ethnicity. Cohort members were distributed almost evenly between urban versus rural locations, and the cohort included Veterans residing in all US Federal Regions and states, with the greatest contribution from region 4 (Southeast US, states of AL, FL, GA, KY, MS, NC, SC, and TN). Overweight (36.3%) and obesity (42.8%) were common among cohort members and 33.6% had a CCI ≥2.

### Trends over time in the associations of risk factors with SARS-CoV-2 infection

During the entire follow-up period (February 1, 2020 to March 31, 2021), 216,046 out of 9,127,673 VA enrollees (2.4%) in our cohort tested positive for SARS-CoV-2 (**Table 1**). Infection rates peaked in accordance with the 3 well-described national waves of the pandemic in April 2020 (first wave), July 2020 (second wave), and December 2020 (third wave) as shown in **Fig 1** (for infection rates), **Fig 2** (for mortality rates), and **Fig 3** (for case fatality rates). Characteristics independently associated with testing positive over the entire period included Black (versus White) race (AOR 1.39, 95% CI 1.37 to 1.41, $p$-value = <0.001), Hispanic ethnicity (AOR 1.64, 95% CI 1.62 to 1.67, $p$-value = <0.001), higher BMI (e.g., BMI 40 versus BMI 25 kg/m$^2$, AOR 1.51, 95% CI 1.49 to 1.53, $p$-value < 0.001), and higher CCI (e.g., CCI ≥ 6 versus CCI = 0 to 1, AOR 3.16, 95% CI 3.12 to 3.21, $p$-value < 0.001) (**Table 2**). The highest risk of infection was observed at age 45 with progressively lower risk at both older ages (e.g., 85 versus 45 years old, AOR 0.50, 95% CI 0.43 to 0.58, $p$-value < 0.001) and younger ages (e.g., 25 versus 45 years old, AOR 0.77, 95% CI 0.76 to 0.79, $p$-value < 0.001). Men had significantly lower risk of infection than women (AOR 0.88, 95% CI 0.87 to 0.90, $p$-value = <0.001). Compared to Federal Region 4 (Southeast US), which served as the reference category, some regions were associated with significantly higher risk of positive SARS-CoV-2 test (e.g., Federal Regions 5 to 9), while others were associated with significantly lower risk (Federal Regions 1 to 3 and 10).

The magnitude of the association between Black (versus White) race and SARS-CoV-2 infection declined steadily from February/March 2020 (AOR 5.10, 95% CI 4.65 to 5.59, $p$-value < 0.001) to November 2020 (AOR 1.03, 95% CI 1.00 to 1.07, $p$-value = 0.05) when it was no longer significant ($p$-value < 0.001 for interaction term of [black race $^*$ time period] testing trends over time, see **S1 and S2 Tables**). However, during the last 4 months of the observation period from December 2020 to March 2021 (corresponding to the third wave of the pandemic), Black race was again significantly associated with infection with AORs ranging from 1.15 to 1.30, although the magnitude of this association was still much lower than in the early months of the pandemic (**Table 2**, **Fig 4**). When we categorized our cohort by age (<65 and

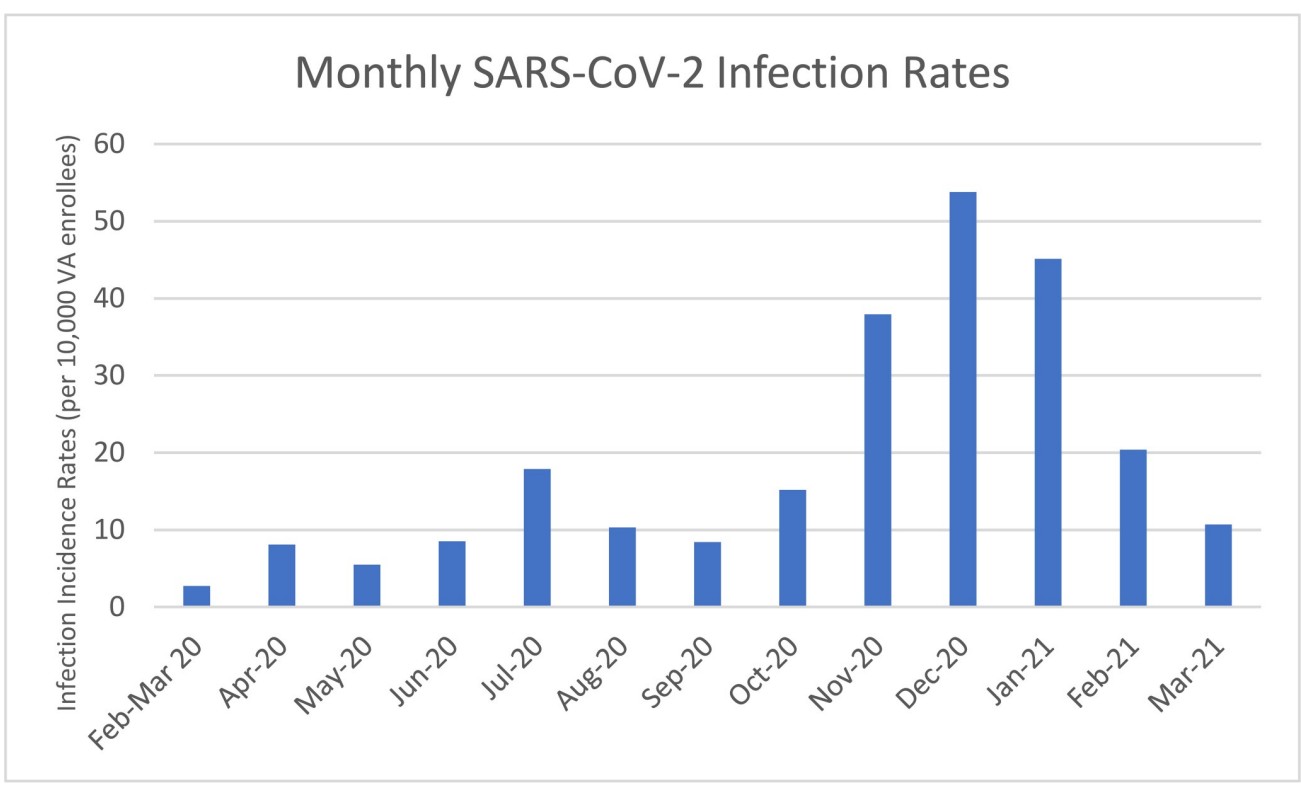

**Fig 1. Monthly results in the VA healthcare system from February 2020 to March 2021 of SARS-CoV-2 infection rates among a cohort of VA enrollees.** SARS-CoV-2, Severe Acute Respiratory Syndrome Coronavirus 2; VA, Veterans Affairs.

≥65 years), the associations and their trends over time between Black (versus White) race and SARS-CoV-2 infection were very similar for persons aged <65 and ≥65 years (**S3 Table**).

The magnitude of the association between AI/AN (versus White) race and SARS-CoV-2 infection declined steadily over time (*p*-value for trends over time <0.001; see **S1 and S2 Tables**) and shifted from a positive association during the period from February to June (AORs ranging from 1.13 to 1.54) to a negative association in March 2021 (AOR 0.66, 95% CI 0.51 to 0.85, *p*-value = 0.004).

The magnitude of the association between urban versus rural location also declined steadily over time (*p*-value for trends over time <0.001; see **S1 and S2 Tables**) and shifted from a positive association in February/March 2020 (AOR 2.02, 95% CI 1.83 to 2.22, *p*-value = <0.001) to a negative association in September to October 2020 and a nonsignificant association in March 2021 (AOR 0.98, 95% CI 0.94 to 1.02, *p*-value = 0.32) (**Table 2**, **Fig 4**).

The magnitude of the associations between CCI and SARS-CoV-2 infection attenuated early in the pandemic until July 2020 (*p*-value < 0.001) and then appeared to plateau between July 2020 and January 2021, before declining again from January to March 2021, after the introduction of vaccination. However, CCI was still strongly associated with infection even in March 2021. Geographical regions at higher risk of infection fluctuated over time reflecting surges in different parts of the country. For example, region 2 (NY, NJ, and PR), which represented the earliest epicenter of the pandemic in the US, had the highest risk of infection in February to March 2020, but one of the lowest risks of infection in July to August 2020 and approximately average risk by March 2021.

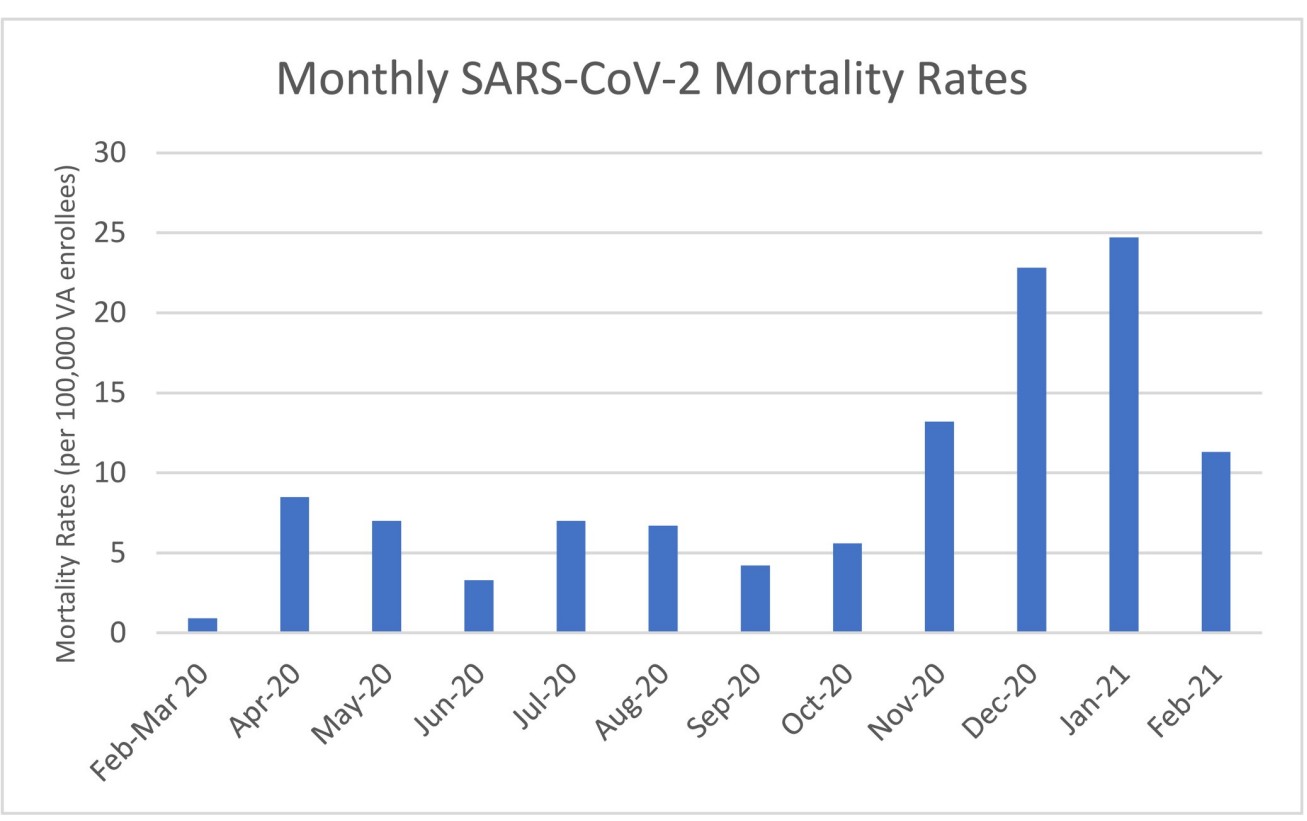

**Fig 2. Monthly results in the VA healthcare system from February 2020 to March 2021 of SARS-CoV-2 mortality rates among a cohort of VA enrollees.** SARS-CoV-2, Severe Acute Respiratory Syndrome Coronavirus 2; VA, Veterans Affairs.

The magnitude of the association of the following characteristics with SARS-CoV-2 infection did not vary appreciably over the observation period: age, sex, PI/NH race, Hispanic ethnicity, and BMI.

## Trends over time in the associations of risk factors with SARS-CoV-2–related mortality

From February 2020 to March 2021, 10,230 SARS-CoV-2–related deaths were identified among our cohort of 9,127,673 VA enrollees (Table 3). Monthly mortality rates are shown in Table 3 and Fig 2. Significant, independent risk factors for mortality over the entire observation period included male sex (AOR 1.59, 95% CI 1.38 to 1.82, $p$-value < 0.001), older age (e.g., AOR 20.37, 95% CI 17.48 to 23.74, $p$-value < 0.001, comparing ages 85 year versus 45 year (the reference age)), Black (AOR 1.55, 95% CI 1.47 to 1.64, $p$-value = <0.001), and AI/AN (AOR 1.66, 95% CI 1.39 to 1.98, $p$-value = <0.001) race relative to White race, Hispanic ethnicity (AOR 1.57, 95% CI 1.45 to 1.70, $p$-value = <0.001), higher BMI (e.g., AOR 1.48, 95% CI 1.40 to 1.57, $p$-value < 0.001, comparing BMI 40 kg/m$^2$ to BMI 25 kg/m$^2$), and higher CCI (AOR 9.46, 95% CI 8.89 to 10.07, $p$-value = <0.001 for CCI $\geq$6 relative to CCI 0 to 1) (Table 4).

The magnitude of the association between Black (versus White) race and SARS-CoV-2–related mortality declined steadily over time from February to April (AOR 3.85, 95% CI 3.30 to 4.50, $p$-value < 0.001) and became nonsignificant by September to October (AOR 1.16, 95% CI 0.95 to 1.43, $p$-value = 0.15) (Table 4, Fig 2); however, Black race was again positively

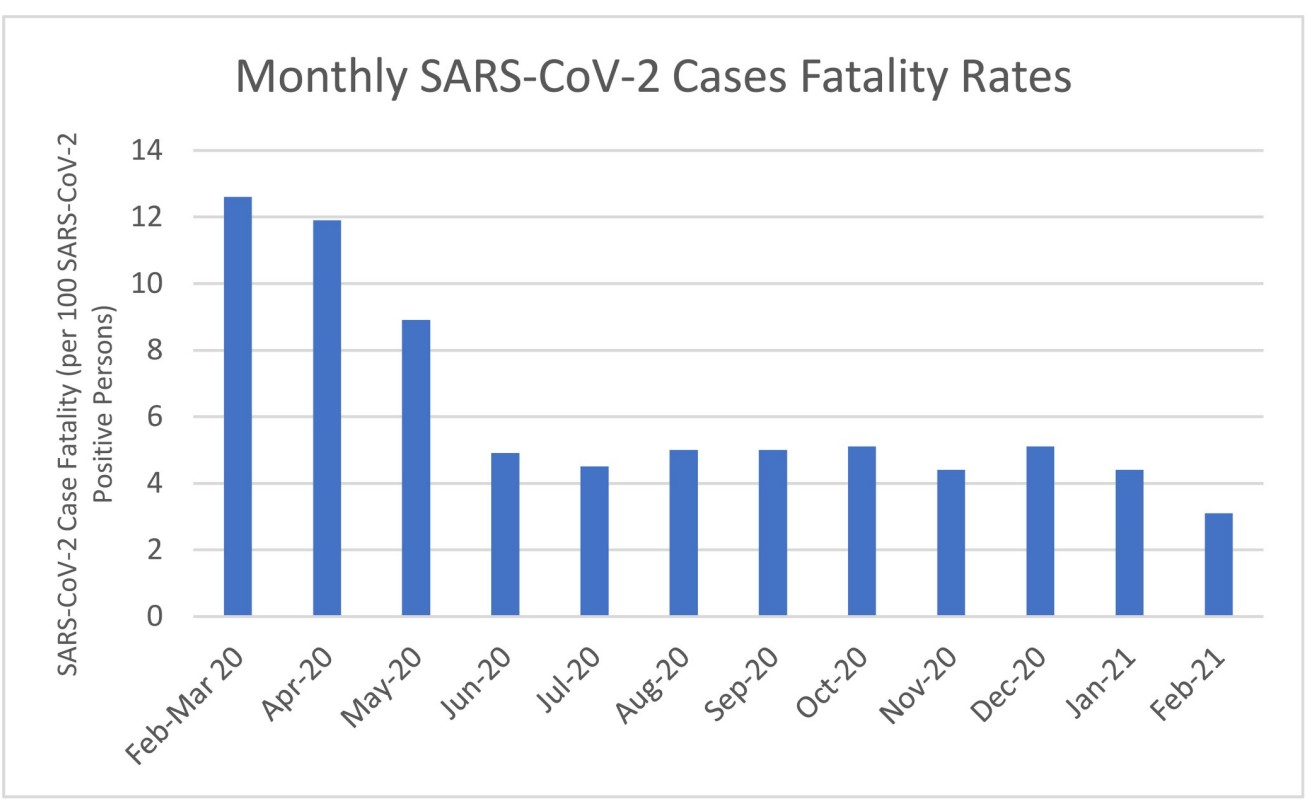

**Fig 3. Monthly results in the VA healthcare system from February 2020 to March 2021 of SARS-CoV-2 case fatality rates among VA enrollees testing positive for SARS-CoV-2.** SARS-CoV-2, Severe Acute Respiratory Syndrome Coronavirus 2; VA, Veterans Affairs.

associated with SARS-CoV-2–related mortality in January to February 2021 (AOR 1.28, 95% CI 1.16 to 1.42, *p*-value = <0.001), albeit at a much lower magnitude of association.

The magnitude of the association between urban versus rural location declined steadily over time (*p*-value for trends over time <0.001, see **S1 and S2 Tables**) and shifted from a positive association in February/April (AOR 2.48, 95% CI 2.08 to 2.96, *p*-value = <0.001) to a negative association in all the time periods after September 2020. Geographical regions with the highest risk of SARS-CoV-2–related mortality in February to April (Federal Regions 1 and 2) had the lowest risk of SARS-CoV-2–related mortality by September to October with some increase in risk thereafter.

Associations between sex, CCI, or age and SARS-CoV-2–related mortality appeared to be stable over time.

### Trends over time in the associations of risk factors with SARS-CoV-2 case fatality

Among a total of 206,789 persons who tested positive for SARS-CoV-2 infection from February 1, 2020 to February 28, 2021, 10,429 (5.0%) SARS-CoV-2–related deaths occurred, i.e., deaths within 30 days of infection (**Table 5**). Case fatality declined progressively over time from 12.6% in persons who tested positive in February to March 2020 to 3.1% in persons who tested positive in February 2021 (**Table 5**, **Fig 3**). Significant, independent risk factors for case fatality over the entire observation period included male sex (AOR 1.73, 95% CI 1.50 to 2.00, *p*-value < 0.001), older age (e.g., AOR 50.09, 95% CI 42.87 to 58.52, *p*-value < 0.001, for age 85

**Table 2. Trends over time in the associations (AORs*) of sociodemographic characteristics and comorbidity burden with risk of SARS-CoV-2 infection in a cohort of 9.1 million VA enrollees from February 2020 to March 2021.**

| | AORs* for SARS-CoV-2 infection (95% CI) | | | | | | | | | | | | | |
|---|---|---|---|---|---|---|---|---|---|---|---|---|---|---|
| | Entire period: February 2020 to March 2021 N = 9,127,673 | February to March 2020 N = 9,127,673 | April 2020 N = 9,090,196 | May 2020 N = 9,053,082 | June 2020 N = 9,022,051 | July 2020 N = 8,990,833 | August 2020 N = 8,948,674 | September 2020 N = 8,913,864 | October 2020 N = 8,881,573 | November 2020 N = 8,844,680 | December 2020 N = 8,804,424 | January 2021 N = 8,744,388 | February 2021 N = 8,671,916 | March 2021 N = 8,630,283 |
| **Sex** | | | | | | | | | | | | | | |
| Female | 1 | 1 | 1 | 1 | 1 | 1 | 1 | 1 | 1 | 1 | 1 | 1 | 1 | 1 |
| Male | 0.88 (0.87 to 0.90) | 1.20 (1.04 to 1.39) | 0.91 (0.84 to 0.98) | 0.94 (0.85 to 1.04) | 0.97 (0.90 to 1.05) | 0.89 (0.85 to 0.94) | 0.92 (0.86 to 0.98) | 0.94 (0.87 to 1.02) | 0.94 (0.88 to 0.99) | 0.87 (0.83 to 0.90) | 0.85 (0.83 to 0.88) | 0.89 (0.86 to 0.92) | 0.86 (0.81 to 0.90) | 0.90 (0.84 to 0.96) |
| **Age (years)** | | | | | | | | | | | | | | |
| 25 | 0.77 (0.76 to 0.79) | 0.62 (0.49 to 0.79) | 0.81 (0.70 to 0.94) | 0.94 (0.79 to 1.12) | 1.25 (1.12 to 1.39) | 0.96 (0.89 to 1.03) | 0.87 (0.78 to 0.97) | 0.76 (0.67 to 0.87) | 0.69 (0.62 to 0.76) | 0.72 (0.68 to 0.77) | 0.75 (0.71 to 0.79) | 0.72 (0.68 to 0.76) | 0.69 (0.63 to 0.76) | 0.71 (0.64 to 0.79) |
| 35 | 0.92 (0.91 to 0.92) | 0.84 (0.76 to 0.93) | 0.92 (0.86 to 0.97) | 0.96 (0.89 to 1.03) | 1.13 (1.08 to 1.18) | 1.01 (0.98 to 1.04) | 0.95 (0.91 to 1.00) | 0.91 (0.86 to 0.96) | 0.87 (0.84 to 0.91) | 0.89 (0.87 to 0.91) | 0.91 (0.89 to 0.93) | 0.89 (0.87 to 0.91) | 0.87 (0.84 to 0.90) | 0.88 (0.84 to 0.92) |
| 45 (reference) | 1 | 1 | 1 | 1 | 1 | 1 | 1 | 1 | 1 | 1 | 1 | 1 | 1 | 1 |
| 55 | 0.89 (0.88 to 0.89) | 0.86 (0.80 to 0.93) | 0.99 (0.94 to 1.04) | 1.09 (1.03 to 1.16) | 0.83 (0.80 to 0.87) | 0.85 (0.82 to 0.88) | 0.93 (0.89 to 0.97) | 0.89 (0.85 to 0.93) | 0.88 (0.85 to 0.91) | 0.86 (0.85 to 0.88) | 0.87 (0.86 to 0.89) | 0.90 (0.88 to 0.92) | 0.94 (0.91 to 0.97) | 0.90 (0.86 to 0.93) |
| 65 | 0.71 (0.70 to 0.73) | 0.65 (0.57 to 0.75) | 0.86 (0.79 to 0.94) | 1.01 (0.91 to 1.13) | 0.65 (0.60 to 0.69) | 0.66 (0.62 to 0.69) | 0.79 (0.73 to 0.85) | 0.73 (0.68 to 0.80) | 0.71 (0.67 to 0.75) | 0.68 (0.65 to 0.71) | 0.70 (0.68 to 0.72) | 0.75 (0.72 to 0.77) | 0.80 (0.76 to 0.85) | 0.68 (0.64 to 0.73) |
| 75 | 0.64 (0.63 to 0.65) | 0.57 (0.51 to 0.65) | 0.75 (0.70 to 0.82) | 0.80 (0.73 to 0.88) | 0.54 (0.50 to 0.58) | 0.54 (0.52 to 0.57) | 0.70 (0.65 to 0.74) | 0.68 (0.64 to 0.73) | 0.66 (0.62 to 0.69) | 0.62 (0.60 to 0.64) | 0.65 (0.63 to 0.67) | 0.71 (0.69 to 0.73) | 0.76 (0.73 to 0.80) | 0.54 (0.51 to 0.58) |
| 85 | 0.58 (0.57 to 0.59) | 0.50 (0.43 to 0.58) | 0.91 (0.84 to 0.99) | 1.03 (0.93 to 1.13) | 0.56 (0.52 to 0.61) | 0.52 (0.49 to 0.55) | 0.67 (0.62 to 0.73) | 0.66 (0.61 to 0.72) | 0.60 (0.57 to 0.64) | 0.57 (0.55 to 0.60) | 0.62 (0.60 to 0.65) | 0.66 (0.64 to 0.69) | 0.71 (0.67 to 0.75) | 0.47 (0.43 to 0.51) |
| **Race** | | | | | | | | | | | | | | |
| White | 1 | 1 | 1 | 1 | 1 | 1 | 1 | 1 | 1 | 1 | 1 | 1 | 1 | 1 |
| Black | 1.39 (1.37 to 1.41) | 5.10 (4.65 to 5.59) | 3.13 (2.97 to 3.31) | 2.55 (2.39 to 2.73) | 1.96 (1.85 to 2.07) | 1.91 (1.84 to 1.98) | 1.75 (1.66 to 1.84) | 1.33 (1.25 to 1.41) | 1.06 (1.01 to 1.11) | 1.03 (1.00 to 1.07) | 1.15 (1.12 to 1.18) | 1.26 (1.23 to 1.29) | 1.30 (1.25 to 1.35) | 1.16 (1.10 to 1.23) |
| Asian | 0.84 (0.80 to 0.87) | 1.78 (1.21 to 2.63) | 1.07 (0.81 to 1.39) | 0.96 (0.69 to 1.34) | 0.70 (0.56 to 0.88) | 0.87 (0.75 to 1.02) | 0.90 (0.72 to 1.13) | 0.99 (0.78 to 1.26) | 0.78 (0.64 to 0.96) | 0.75 (0.67 to 0.85) | 0.82 (0.75 to 0.89) | 0.84 (0.76 to 0.93) | 0.89 (0.77 to 1.03) | 0.74 (0.59 to 0.92) |
| AI/AN | 0.93 (0.89 to 0.98) | 1.13 (0.67 to 1.92) | 1.40 (1.07 to 1.83) | 1.54 (1.16 to 2.05) | 1.21 (0.98 to 1.48) | 0.94 (0.80 to 1.11) | 0.89 (0.71 to 1.11) | 1.00 (0.79 to 1.25) | 0.99 (0.84 to 1.04) | 0.93 (0.84 to 1.04) | 0.88 (0.80 to 0.97) | 0.89 (0.80 to 0.99) | 0.97 (0.83 to 1.13) | 0.66 (0.51 to 0.85) |
| PI/NH | 0.98 (0.93 to 1.02) | 0.99 (0.57 to 1.70) | 1.31 (1.01 to 1.71) | 1.34 (1.00 to 1.81) | 0.95 (0.75 to 1.19) | 0.95 (0.81 to 1.11) | 1.22 (1.00 to 1.49) | 1.12 (0.89 to 1.41) | 0.97 (0.81 to 1.17) | 0.97 (0.87 to 1.09) | 0.88 (0.80 to 0.97) | 0.98 (0.89 to 1.08) | 1.06 (0.91 to 1.23) | 0.78 (0.62 to 1.00) |
| Missing/unknown/refused | 0.82 (0.81 to 0.84) | 1.27 (1.05 to 1.55) | 1.05 (0.93 to 1.17) | 1.10 (0.96 to 1.25) | 0.84 (0.77 to 0.93) | 0.89 (0.83 to 0.95) | 0.85 (0.77 to 0.93) | 0.87 (0.78 to 0.96) | 0.79 (0.73 to 0.86) | 0.81 (0.77 to 0.85) | 0.81 (0.77 to 0.84) | 0.80 (0.76 to 0.84) | 0.78 (0.73 to 0.84) | 0.77 (0.70 to 0.85) |
| **Ethnicity** | | | | | | | | | | | | | | |
| Non-Hispanic | 1 | 1 | 1 | 1 | 1 | 1 | 1 | 1 | 1 | 1 | 1 | 1 | 1 | 1 |
| Hispanic | 1.64 (1.62 to 1.67) | 1.57 (1.37 to 1.81) | 1.27 (1.16 to 1.38) | 1.55 (1.40 to 1.73) | 2.39 (2.24 to 2.56) | 2.55 (2.43 to 2.67) | 2.10 (1.96 to 2.25) | 1.57 (1.44 to 1.71) | 1.57 (1.47 to 1.67) | 1.46 (1.40 to 1.52) | 1.53 (1.48 to 1.58) | 1.49 (1.43 to 1.54) | 1.38 (1.30 to 1.46) | 1.48 (1.37 to 1.60) |
| Missing/unknown/refused | 0.40 (0.39 to 0.41) | 0.38 (0.29 to 0.50) | 0.37 (0.32 to 0.43) | 0.32 (0.27 to 0.39) | 0.44 (0.38 to 0.50) | 0.41 (0.37 to 0.45) | 0.39 (0.34 to 0.45) | 0.36 (0.31 to 0.41) | 0.38 (0.34 to 0.42) | 0.37 (0.34 to 0.39) | 0.41 (0.39 to 0.43) | 0.43 (0.40 to 0.46) | 0.42 (0.38 to 0.46) | 0.40 (0.35 to 0.46) |
| **US Federal Region†** | | | | | | | | | | | | | | |
| 1 | 0.89 (0.86 to 0.91) | 2.48 (1.97 to 3.12) | 6.04 (5.50 to 6.62) | 2.89 (2.56 to 3.25) | 0.57 (0.49 to 0.66) | 0.17 (0.15 to 0.21) | 0.21 (0.18 to 0.26) | 0.28 (0.23 to 0.35) | 0.46 (0.40 to 0.52) | 1.07 (1.00 to 1.15) | 0.97 (0.92 to 1.02) | 0.86 (0.81 to 0.91) | 0.92 (0.86 to 1.00) | 1.12 (1.00 to 1.24) |
| 2 | 0.72 (0.71 to 0.74) | 5.81 (5.08 to 6.65) | 5.14 (4.75 to 5.56) | 1.85 (1.66 to 2.05) | 0.33 (0.29 to 0.38) | 0.18 (0.16 to 0.20) | 0.22 (0.19 to 0.25) | 0.32 (0.28 to 0.37) | 0.44 (0.39 to 0.49) | 0.79 (0.74 to 0.84) | 0.69 (0.65 to 0.72) | 0.64 (0.61 to 0.68) | 0.72 (0.67 to 0.77) | 1.10 (1.01 to 1.20) |
| 3 | 0.83 (0.81 to 0.84) | 1.22 (1.03 to 1.46) | 1.84 (1.68 to 2.01) | 1.59 (1.44 to 1.76) | 0.47 (0.42 to 0.51) | 0.33 (0.30 to 0.35) | 0.44 (0.40 to 0.48) | 0.52 (0.47 to 0.57) | 0.69 (0.64 to 0.74) | 1.14 (1.09 to 1.19) | 1.01 (0.97 to 1.05) | 0.81 (0.78 to 0.84) | 0.85 (0.80 to 0.90) | 1.06 (0.99 to 1.14) |
| 4 (reference) | 1 | 1 | 1 | 1 | 1 | 1 | 1 | 1 | 1 | 1 | 1 | 1 | 1 | 1 |
| 5 | 1.13 (1.11 to 1.15) | 2.50 (2.17 to 2.88) | 2.25 (2.07 to 2.45) | 1.86 (1.70 to 2.03) | 0.48 (0.44 to 0.53) | 0.40 (0.38 to 0.43) | 0.57 (0.53 to 0.62) | 0.87 (0.81 to 0.93) | 1.79 (1.70 to 1.89) | 2.72 (2.63 to 2.82) | 1.23 (1.20 to 1.27) | 0.73 (0.71 to 0.76) | 0.75 (0.71 to 0.79) | 1.07 (1.01 to 1.15) |
| 6 | 1.13 (1.12 to 1.15) | 2.76 (2.41 to 3.17) | 1.30 (1.18 to 1.43) | 1.00 (0.89 to 1.11) | 1.30 (1.22 to 1.38) | 1.23 (1.18 to 1.28) | 0.93 (0.88 to 0.99) | 0.99 (0.92 to 1.06) | 1.33 (1.26 to 1.41) | 1.50 (1.44 to 1.56) | 1.10 (1.07 to 1.13) | 1.04 (1.01 to 1.07) | 0.86 (0.81 to 0.90) | 0.85 (0.79 to 0.92) |
| 7 | 1.42 (1.39 to 1.44) | 1.22 (0.93 to 1.62) | 1.46 (1.26 to 1.68) | 1.56 (1.35 to 1.79) | 0.54 (0.47 to 0.62) | 0.60 (0.55 to 0.66) | 1.12 (1.03 to 1.22) | 1.92 (1.77 to 2.08) | 2.38 (2.23 to 2.53) | 3.37 (3.23 to 3.52) | 1.44 (1.38 to 1.50) | 0.93 (0.88 to 0.97) | 0.93 (0.87 to 1.00) | 1.00 (0.91 to 1.11) |
| 8 | 1.17 (1.14 to 1.19) | 2.65 (2.06 to 3.42) | 2.04 (1.76 to 2.37) | 1.67 (1.43 to 1.97) | 0.48 (0.40 to 0.56) | 0.36 (0.32 to 0.41) | 0.53 (0.46 to 0.60) | 1.23 (1.11 to 1.37) | 2.39 (2.22 to 2.56) | 3.10 (2.95 to 3.25) | 1.18 (1.12 to 1.24) | 0.61 (0.57 to 0.65) | 0.68 (0.62 to 0.74) | 1.03 (0.92 to 1.15) |
| 9 | 1.23 (1.21 to 1.24) | 1.36 (1.12 to 1.64) | 0.99 (0.88 to 1.12) | 0.90 (0.79 to 1.02) | 1.31 (1.22 to 1.40) | 0.93 (0.88 to 0.98) | 0.71 (0.66 to 0.77) | 0.68 (0.62 to 0.74) | 0.69 (0.64 to 0.75) | 1.37 (1.31 to 1.43) | 1.73 (1.68 to 1.78) | 1.34 (1.30 to 1.38) | 1.08 (1.03 to 1.14) | 0.89 (0.82 to 0.96) |
| 10 | 0.68 (0.66 to 0.70) | 1.38 (1.00 to 1.89) | 1.16 (0.97 to 1.40) | 0.79 (0.64 to 0.98) | 0.44 (0.37 to 0.52) | 0.48 (0.43 to 0.53) | 0.60 (0.53 to 0.68) | 0.62 (0.54 to 0.71) | 0.89 (0.81 to 0.99) | 1.23 (1.15 to 1.31) | 0.73 (0.68 to 0.77) | 0.43 (0.40 to 0.47) | 0.56 (0.51 to 0.62) | 0.80 (0.71 to 0.90) |
| **Urban versus rural** | | | | | | | | | | | | | | |

*(Continued)*

**Table 2.** (Continued)

| | AORs[†] for SARS-CoV-2 infection (95% CI) | | | | | | | | | | | | | |
|---|---|---|---|---|---|---|---|---|---|---|---|---|---|---|
| | Entire period: February 2020 to March 2021 N = 9,127,673 | February to March 2020 N = 9,127,673 | April 2020 N = 9,090,196 | May 2020 N = 9,053,082 | June 2020 N = 9,022,051 | July 2020 N = 8,990,833 | August 2020 N = 8,948,674 | September 2020 N = 8,913,864 | October 2020 N = 8,881,573 | November 2020 N = 8,844,680 | December 2020 N = 8,804,424 | January 2021 N = 8,744,388 | February 2021 N = 8,671,916 | March 2021 N = 8,630,283 |
| Rural | 1 | 1 | 1 | 1 | 1 | 1 | 1 | 1 | 1 | 1 | 1 | 1 | 1 | 1 |
| Urban | 1.01 (1.00 to 1.02) | 2.02 (1.83 to 2.22) | 1.95 (1.85 to 2.06) | 1.60 (1.50 to 1.70) | 1.39 (1.32 to 1.45) | 1.23 (1.20 to 1.28) | 0.93 (0.89 to 0.97) | 0.85 (0.81 to 0.89) | 0.88 (0.85 to 0.91) | 0.96 (0.94 to 0.98) | 0.94 (0.92 to 0.96) | 0.96 (0.94 to 0.98) | 0.96 (0.93 to 0.99) | 0.98 (0.94 to 1.02) |
| **BMI (kg/m²)** | | | | | | | | | | | | | | |
| 15 | | | | | | | | | | | | | | |
| 20 | 0.99 (0.97 to 1.00) | 1.12 (0.96 to 1.31) | 1.40 (1.29 to 1.52) | 1.49 (1.35 to 1.63) | 1.18 (1.08 to 1.30) | 0.97 (0.90 to 1.04) | 0.99 (0.90 to 1.08) | 1.03 (0.93 to 1.14) | 0.94 (0.87 to 1.02) | 0.88 (0.83 to 0.92) | 0.98 (0.94 to 1.02) | 1.01 (0.97 to 1.06) | 1.04 (0.97 to 1.11) | 1.15 (1.05 to 1.26) |
| 25 (reference) | 1 | 1 | 1 | 1 | 1 | 1 | 1 | 1 | 1 | 1 | 1 | 1 | 1 | 1 |
| 30 | 1.23 (1.21 to 1.25) | 1.21 (1.06 to 1.37) | 1.14 (1.06 to 1.23) | 1.10 (1.01 to 1.20) | 1.24 (1.15 to 1.34) | 1.19 (1.13 to 1.25) | 1.14 (1.06 to 1.22) | 1.17 (1.09 to 1.26) | 1.28 (1.21 to 1.36) | 1.22 (1.17 to 1.26) | 1.24 (1.20 to 1.28) | 1.24 (1.20 to 1.28) | 1.22 (1.16 to 1.28) | 1.27 (1.19 to 1.36) |
| 35 | 1.38 (1.36 to 1.39) | 1.36 (1.21 to 1.52) | 1.26 (1.18 to 1.34) | 1.14 (1.05 to 1.24) | 1.28 (1.20 to 1.37) | 1.32 (1.26 to 1.38) | 1.36 (1.28 to 1.44) | 1.28 (1.20 to 1.36) | 1.41 (1.34 to 1.48) | 1.39 (1.35 to 1.44) | 1.39 (1.35 to 1.43) | 1.37 (1.33 to 1.41) | 1.36 (1.30 to 1.42) | 1.43 (1.35 to 1.52) |
| 40 | 1.51 (1.49 to 1.53) | 1.41 (1.26 to 1.58) | 1.39 (1.30 to 1.48) | 1.22 (1.12 to 1.32) | 1.39 (1.30 to 1.49) | 1.46 (1.40 to 1.53) | 1.45 (1.37 to 1.54) | 1.38 (1.29 to 1.47) | 1.54 (1.47 to 1.62) | 1.52 (1.47 to 1.57) | 1.52 (1.48 to 1.57) | 1.50 (1.46 to 1.55) | 1.50 (1.44 to 1.57) | 1.59 (1.50 to 1.69) |
| **CCI** | | | | | | | | | | | | | | |
| 0 to 1 | 1 | 1 | 1 | 1 | 1 | 1 | 1 | 1 | 1 | 1 | 1 | 1 | 1 | 1 |
| 2 to 3 | 2.16 (2.14 to 2.19) | 2.25 (2.01 to 2.52) | 2.24 (2.10 to 2.40) | 2.28 (2.10 to 2.47) | 1.98 (1.85 to 2.11) | 2.14 (2.05 to 2.23) | 2.01 (1.90 to 2.13) | 2.18 (2.05 to 2.32) | 2.12 (2.02 to 2.22) | 2.09 (2.03 to 2.15) | 2.20 (2.14 to 2.25) | 2.25 (2.19 to 2.31) | 2.10 (2.02 to 2.18) | 2.00 (1.89 to 2.11) |
| 4 to 5 | 2.56 (2.52 to 2.60) | 2.80 (2.45 to 3.20) | 3.04 (2.82 to 3.28) | 3.10 (2.83 to 3.40) | 2.61 (2.41 to 2.82) | 2.42 (2.29 to 2.55) | 2.53 (2.37 to 2.71) | 2.61 (2.42 to 2.82) | 2.40 (2.27 to 2.54) | 2.49 (2.40 to 2.58) | 2.62 (2.54 to 2.70) | 2.67 (2.59 to 2.76) | 2.38 (2.27 to 2.50) | 2.11 (1.97 to 2.26) |
| ≥6 | 3.16 (3.12 to 3.21) | 4.53 (4.03 to 5.10) | 4.68 (4.37 to 5.01) | 4.34 (3.99 to 4.72) | 3.37 (3.13 to 3.62) | 3.08 (2.93 to 3.25) | 3.16 (2.96 to 3.38) | 3.33 (3.10 to 3.58) | 3.22 (3.06 to 3.40) | 3.02 (2.92 to 3.13) | 3.31 (3.21 to 3.41) | 3.20 (3.10 to 3.30) | 2.95 (2.81 to 3.09) | 2.51 (2.35 to 2.69) |

* Adjusted for sex, age (modeled as restricted cubic splines with 5 knots at ages 30, 49, 64, 73, and 88 years), race, ethnicity, geographical region, urban/rural location, BMI (modeled as restricted cubic splines with 5 knots at BMIs of 21.3, 26.0, 29.0, 32.5, 39.9 kg/m²), and CCI.

[†] Categorized according to the 10 Federal Regions drawn up by the Federal Emergency Management Agency: 1 (CT, MA, ME, NH, RI, and VT), 2 (NJ, NY, and PR), 3 (DC, DE, MD, PA, VA, and WV), 4 (AL, FL, GA, KY, MS, NC, SC, and TN), 5 (IL, IN, MI, MN, OH, and WI), 6 (AR, LA, NM, OK, and TX), 7 (IA, KS, MO, and NE), 8 (CO, MT, ND, SD, UT, and WY), 9 (AZ, CA, GU, HI, and NV), and 10 (AK, ID, OR, and WA).

AI/AN, American Indian/Alaska Native; AOR, adjusted odds ratio; BMI, body mass index; CCI, Charlson comorbidity index; PI/NH, Pacific Islander/Native Hawaiian; SARS-CoV-2, Severe Acute Respiratory Syndrome Coronavirus 2; VA, Veterans Affairs.

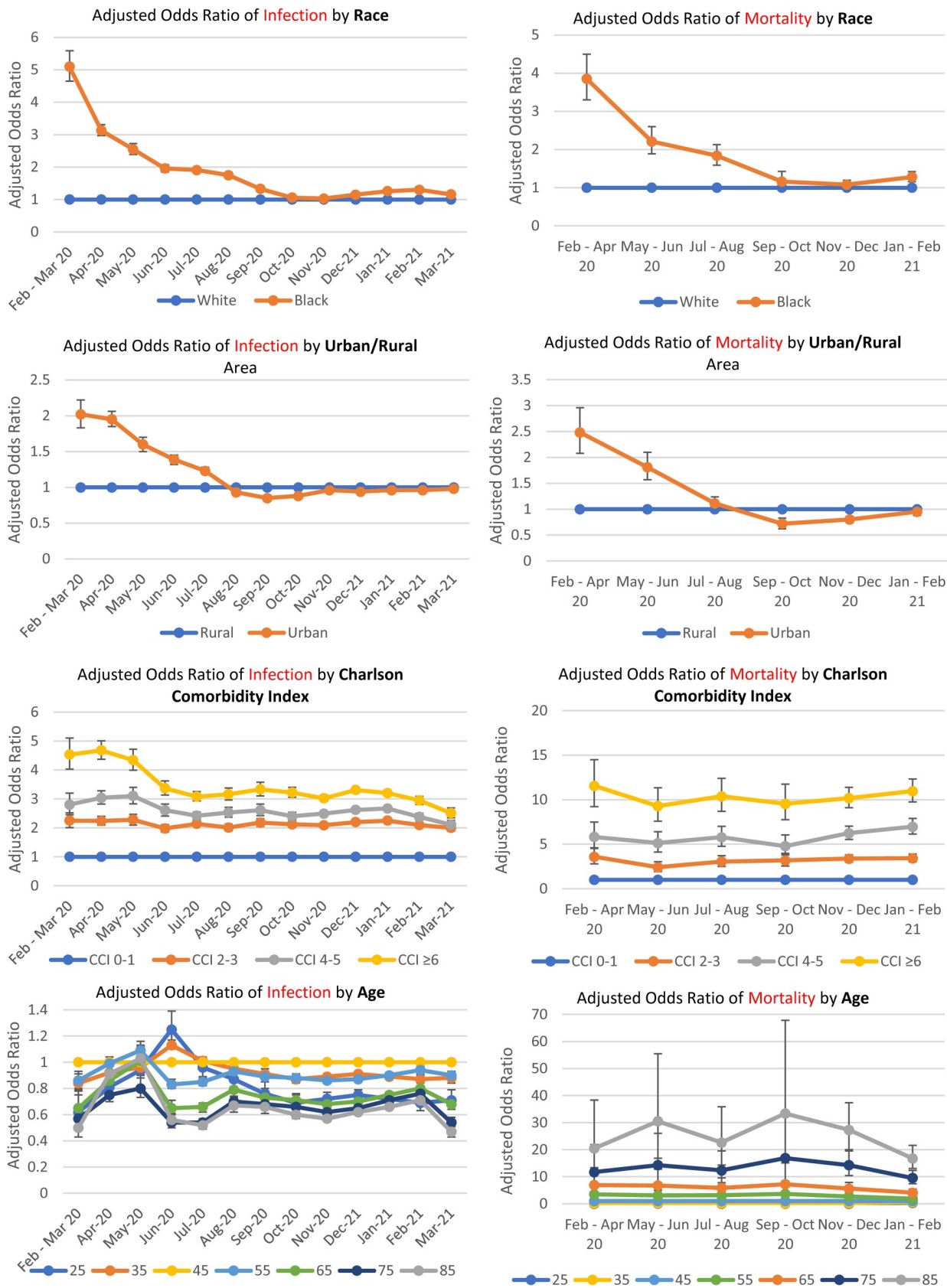

**Fig 4.** Trends over time in the associations of the following factors with risk of SARS-CoV-2 infection and mortality: **(A and B)** Black versus White race. **(C and D)** Urban versus rural location. **(E and F)** CCI categories. **(G and H)** Age. CCI, Charlson comorbidity index; SARS-CoV-2, Severe Acute Respiratory Syndrome Coronavirus 2.

versus age 45 years), Black (AOR 1.13, 95% CI 1.06 to 1.19, $p$-value < 0.001), and AI/AN (AOR 1.76, 95% CI 1.44 to 2.14, $p$-value < 0.001) race relative to White race, Hispanic versus non-Hispanic ethnicity (AOR 1.21, 95% CI 1.11 to 1.31, $p$-value < 0.001), obesity/high BMI (e.g., AOR 1.35, 95% CI 1.27 to 1.43, $p$-value < 0.001 comparing BMI 40 versus 25 kg/m$^2$) or underweight/low BMI (e.g., AOR 1.44, 95% CI 1.35 to 1.53, $p$-value < 0.001 comparing BMI 20 versus 25 kg/m$^2$), and progressively higher CCI (e.g., AOR 2.51, 95% CI 2.35 to 2.68, $p$-value = <0.001 for CCI ≥6 relative to CCI 0 to 1) (Table 6).

The magnitude of the association between Black (versus White) race and case fatality declined over time and shifted from a positive association in February to April (AOR 2.56, 95% CI 2.23 to 2.93, $p$-value < 0.001) to a negative association in all time periods after September 2020 (Table 6). Also, the magnitude of the association between urban versus rural location and case fatality declined steadily over time and shifted from a positive association in February/April (AOR 2.24, 95% CI 1.93 to 2.60, $p$-value = <0.001) to a negative association in all the time periods after September 2020. Geographical regions with some of the highest case fatality rates in February to April (e.g., Federal Regions 1, 2, and 5) had some of the lowest case fatality rates by January/February 2021. Associations between sex, CCI, or age and case fatality appeared to be stable over time.

## Discussion

Our study of a national cohort of 9.1 million VA enrollees followed from February 2020 to March 2021 demonstrates that the strongly positive associations between Black (versus White) race and SARS-CoV-2 infection, mortality, and case fatality that were observed in the early months of the pandemic attenuated over time and were no longer significant by November 2020. Positive associations between AI/AN (versus White) race and risk of infection noted early in the pandemic also attenuated over time and reversed by March 2021. Similarly, strongly positive associations between urban (versus rural) location and SARS-CoV-2 infection, mortality, and case fatality that were present early in the pandemic attenuated over time and were no longer significant by March 2021. Throughout the observation period, high comorbidity burden, younger age, Hispanic ethnicity, and obesity were consistently associated with infection, while high comorbidity burden, older age, Hispanic ethnicity, obesity, and male sex were consistently associated with mortality.

Multiple studies reported higher risk of SARS-CoV-2 infection and mortality in Black versus White persons [8,9,14]. In a nationally representative US study, Black persons accounted for 18.7% of SARS-CoV-2–related deaths from May to August 2020 despite representing just 12.5% of the US population. During this time period, the percentage of decedents who were Black decreased from 20.3% in May to 17.4% in August but was still higher than the percentage of Black persons in the US population [8]. Our study extends these prior findings in significant ways. First, we demonstrate that the association between Black race and SARS-CoV-2–related mortality continued to decline in the VA healthcare system even after August and became nonsignificant by November 2020. Second, we simultaneously adjusted for comorbidity burden, age, sex, ethnicity, BMI, geographic region, and rural/urban location to identify the associations of Black race and mortality that were not explained by differences in these factors. Third, we show that the associations of Black race with infection also declined over time and became nonsignificant by November 2020, while the positive associations with case fatality

**Table 3. SARS-CoV-2–related mortality presented by month in a cohort of 9.1 million VA enrollees followed from February 2020 to February 2021.**

| Time period and number of VA enrollees at risk* | Entire period: February 2020 to February 2021 N = 9,127,673 | February to March 2020 N = 9,127,673 | April 2020 N = 9,090,196 | May 2020 N = 9,053,082 | June 2020 N = 9,022,051 | July 2020 N = 8,990,833 | August 2020 N = 8,948,674 | September 2020 N = 8,913,864 | October 2020 N = 8,881,573 | November 2020 N = 8,844,680 | December 2020 N = 8,804,424 | January 2021 N = 8,744,388 | February 2021 N = 8,671,916 |
|---|---|---|---|---|---|---|---|---|---|---|---|---|---|
| **All persons** | 10,230 (9.3) | 83 (0.9) | 775 (8.5) | 631 (7.0) | 295 (3.3) | 627 (7.0) | 604 (6.7) | 375 (4.2) | 500 (5.6) | 1,165 (13.2) | 2,019 (22.8) | 2,172 (24.7) | 984 (11.3) |
| **Sex** | | | | | | | | | | | | | |
| Female | 208 (2.1) | 1 (0.1) | 14 (1.7) | 18 (2.2) | 7 (0.8) | 14 (1.7) | 15 (1.8) | 3 (0.4) | 11 (1.3) | 21 (2.6) | 42 (5.1) | 38 (4.7) | 24 (3.0) |
| Male | 10,022 (10.1) | 82 (1.0) | 761 (9.2) | 613 (7.4) | 288 (3.5) | 613 (7.5) | 589 (7.2) | 372 (4.6) | 489 (6.1) | 1,144 (14.2) | 1,977 (24.7) | 2,134 (26.8) | 960 (12.1) |
| **Age (years)** | | | | | | | | | | | | | |
| 18 to 24 | 0 (0.0) | 0 (0.0) | 0 (0.0) | 0 (0.0) | 0 (0.0) | 0 (0.0) | 0 (0.0) | 0 (0.0) | 0 (0.0) | 0 (0.0) | 0 (0.0) | 0 (0.0) | 0 (0.0) |
| 25 to 34 | 10 (0.1) | 0 (0.0) | 0 (0.0) | 0 (0.0) | 0 (0.0) | 0 (0.0) | 0 (0.0) | 0 (0.0) | 0 (0.0) | 0 (0.0) | 2 (0.2) | 3 (0.4) | 5 (0.6) |
| 35 to 44 | 40 (0.3) | 0 (0.0) | 1 (0.1) | 3 (0.3) | 0 (0.0) | 5 (0.5) | 0 (0.0) | 1 (0.1) | 1 (0.1) | 1 (0.1) | 7 (0.7) | 13 (1.2) | 8 (0.8) |
| 45 to 54 | 153 (1.1) | 4 (0.3) | 16 (1.4) | 8 (0.7) | 4 (0.3) | 10 (0.9) | 12 (1.0) | 7 (0.6) | 10 (0.9) | 9 (0.8) | 34 (2.9) | 28 (2.4) | 11 (1.0) |
| 55 to 64 | 743 (4.1) | 11 (0.7) | 71 (4.7) | 36 (2.4) | 26 (1.7) | 59 (3.9) | 54 (3.6) | 21 (1.4) | 30 (2.0) | 68 (4.6) | 134 (9.0) | 160 (10.8) | 73 (5.0) |
| 65 to 74 | 3,354 (11.5) | 28 (1.1) | 272 (11.2) | 181 (7.5) | 86 (3.6) | 213 (8.9) | 206 (8.6) | 125 (5.2) | 165 (6.9) | 363 (15.3) | 662 (28.0) | 733 (31.2) | 320 (13.8) |
| 75 to 84 | 2,993 (19.8) | 20 (1.6) | 180 (14.4) | 157 (12.7) | 77 (6.2) | 171 (14.0) | 162 (13.3) | 113 (9.3) | 148 (12.3) | 366 (30.7) | 629 (52.8) | 656 (55.5) | 314 (26.9) |
| ≥85 | 2,937 (32.0) | 20 (2.6) | 235 (31.3) | 246 (33.4) | 102 (14.0) | 169 (23.6) | 170 (24.1) | 108 (15.5) | 146 (21.3) | 358 (52.9) | 551 (81.9) | 579 (86.9) | 253 (38.9) |
| **Race** | | | | | | | | | | | | | |
| White | 7,273 (10.3) | 34 (0.6) | 411 (7.0) | 406 (7.0) | 194 (3.3) | 409 (7.1) | 396 (6.9) | 271 (4.7) | 398 (7.0) | 913 (16.0) | 1,541 (27.1) | 1,561 (27.6) | 739 (13.2) |
| Black | 1,935 (12.1) | 44 (3.3) | 309 (23.2) | 174 (13.1) | 74 (5.6) | 150 (11.4) | 138 (10.5) | 62 (4.7) | 59 (4.5) | 112 (8.6) | 277 (21.4) | 374 (29.0) | 162 (12.7) |
| Asian | 65 (4.7) | 0 (0.0) | 3 (2.6) | 6 (5.2) | 2 (1.8) | 3 (2.6) | 8 (7.0) | 5 (4.4) | 1 (0.9) | 2 (1.8) | 15 (13.3) | 15 (13.3) | 5 (4.5) |
| AI/AN | 125 (13.1) | 0 (0.0) | 6 (7.5) | 5 (6.3) | 4 (5.1) | 9 (11.4) | 8 (10.2) | 7 (9.0) | 9 (11.6) | 14 (18.0) | 27 (34.9) | 28 (36.4) | 8 (10.5) |
| PI/NH | 87 (9.6) | 1 (1.3) | 4 (5.3) | 3 (4.0) | 3 (4.0) | 6 (8.1) | 4 (5.4) | 2 (2.7) | 3 (4.1) | 12 (16.4) | 20 (27.3) | 21 (28.9) | 8 (11.1) |
| Missing/unknown/refused | 745 (3.8) | 4 (0.2) | 42 (2.6) | 37 (2.3) | 18 (1.1) | 50 (3.1) | 50 (3.1) | 28 (1.7) | 30 (1.9) | 112 (7.0) | 139 (8.7) | 173 (10.9) | 62 (3.9) |
| **Ethnicity** | | | | | | | | | | | | | |
| Non-Hispanic | 9,097 (10.5) | 75 (1.0) | 711 (9.9) | 577 (8.1) | 257 (3.6) | 516 (7.3) | 529 (7.5) | 341 (4.8) | 458 (6.5) | 1,053 (15.1) | 1,813 (26.1) | 1,904 (27.5) | 863 (12.6) |
| Hispanic | 745 (10.9) | 8 (1.4) | 45 (7.9) | 32 (5.6) | 28 (4.9) | 89 (15.7) | 51 (9.0) | 21 (3.7) | 23 (4.1) | 75 (13.5) | 133 (23.9) | 158 (28.6) | 82 (15.0) |
| Missing/unknown/refused | 388 (2.4) | 0 (0.0) | 19 (1.4) | 22 (1.6) | 10 (0.7) | 22 (1.6) | 24 (1.8) | 13 (1.0) | 19 (1.4) | 37 (2.8) | 73 (5.5) | 110 (8.3) | 39 (3.0) |
| **US Federal Region†** | | | | | | | | | | | | | |
| 1 | 418 (8.7) | 2 (0.5) | 99 (24.8) | 86 (21.7) | 13 (3.3) | 6 (1.5) | 6 (1.5) | 3 (0.8) | 3 (0.8) | 24 (6.2) | 70 (18.1) | 65 (16.9) | 41 (10.7) |
| 2 | 704 (8.8) | 24 (3.6) | 237 (35.8) | 115 (17.5) | 16 (2.5) | 13 (2.0) | 4 (0.6) | 5 (0.8) | 9 (1.4) | 25 (3.9) | 82 (12.8) | 114 (17.9) | 60 (9.5) |
| 3 | 905 (7.9) | 3 (0.3) | 86 (9.0) | 72 (7.6) | 36 (3.8) | 29 (3.1) | 31 (3.3) | 19 (2.0) | 46 (4.9) | 80 (8.6) | 190 (20.4) | 213 (23.0) | 100 (10.9) |
| 4 | 2,385 (8.7) | 7 (0.3) | 77 (3.4) | 106 (4.7) | 85 (3.7) | 209 (9.2) | 237 (10.5) | 145 (6.5) | 116 (5.2) | 181 (8.1) | 362 (16.3) | 590 (26.7) | 270 (12.3) |
| 5 | 1,567 (9.8) | 17 (1.3) | 134 (10.2) | 113 (8.6) | 41 (3.1) | 34 (2.6) | 56 (4.3) | 35 (2.7) | 78 (6.0) | 316 (24.6) | 387 (30.2) | 236 (18.6) | 120 (9.5) |
| 6 | 1,567 (10.9) | 20 (1.7) | 74 (6.2) | 48 (4.0) | 33 (2.8) | 179 (15.2) | 133 (11.3) | 53 (4.5) | 91 (7.8) | 212 (18.3) | 273 (23.6) | 306 (26.6) | 145 (12.7) |
| 7 | 710 (12.9) | 2 (0.4) | 16 (3.5) | 28 (6.2) | 9 (2.0) | 12 (2.7) | 35 (7.8) | 53 (11.8) | 67 (15.0) | 143 (32.2) | 170 (38.5) | 125 (28.6) | 50 (11.6) |
| 8 | 387 (8.6) | 2 (0.5) | 11 (2.9) | 24 (6.5) | 11 (3.0) | 8 (2.2) | 7 (1.9) | 19 (5.2) | 48 (13.1) | 83 (22.7) | 104 (28.6) | 51 (14.1) | 19 (5.3) |
| 9 | 1,340 (10.7) | 3 (0.3) | 33 (3.2) | 26 (2.5) | 46 (4.5) | 123 (12.0) | 76 (7.4) | 30 (2.9) | 28 (2.8) | 68 (6.7) | 325 (32.1) | 425 (42.2) | 157 (15.8) |
| 10 | 247 (4.9) | 3 (0.7) | 8 (1.9) | 13 (3.1) | 5 (1.2) | 14 (3.4) | 19 (4.6) | 13 (3.2) | 14 (3.4) | 33 (8.1) | 56 (13.7) | 47 (11.6) | 22 (5.4) |
| **Urban versus rural** | | | | | | | | | | | | | |
| Rural | 5,002 (9.3) | 20 (0.4) | 151 (3.4) | 171 (3.9) | 116 (2.6) | 263 (6.0) | 319 (7.3) | 226 (5.2) | 320 (7.3) | 707 (16.3) | 1,124 (26.0) | 1,104 (25.6) | 481 (11.3) |
| Urban | 5,228 (9.4) | 63 (1.4) | 624 (13.4) | 460 (10.0) | 179 (3.9) | 364 (7.9) | 285 (6.2) | 149 (3.3) | 180 (4.0) | 458 (10.1) | 895 (19.9) | 1,068 (23.8) | 503 (11.3) |
| **BMI (kg/m²)** | | | | | | | | | | | | | |

(Continued)

**Table 3.** (Continued)

| Time period and number of VA enrollees at risk* | Number of SARS-CoV-2–related deaths per month (and mortality per 100,000 persons per month) | | | | | | | | | | | | |
|---|---|---|---|---|---|---|---|---|---|---|---|---|---|
| | Entire period: February 2020 to February 2021 N = 9,127,673 | February to March 2020 N = 9,127,673 | April 2020 N = 9,090,196 | May 2020 N = 9,053,082 | June 2020 N = 9,022,051 | July 2020 N = 8,990,833 | August 2020 N = 8,948,674 | September 2020 N = 8,913,864 | October 2020 N = 8,881,573 | November 2020 N = 8,844,680 | December 2020 N = 8,804,424 | January 2021 N = 8,744,388 | February 2021 N = 8,671,916 |
| <18.5 (underweight) | 304 (39.2) | 3 (4.6) | 28 (44.4) | 28 (45.2) | 16 (26.3) | 19 (31.8) | 15 (25.5) | 10 (17.3) | 12 (21.0) | 32 (56.7) | 46 (82.0) | 66 (118.8) | 29 (53.2) |
| 18.5 to <25 (normal weight) | 2,660 (12.0) | 21 (1.1) | 222 (12.1) | 215 (11.8) | 83 (4.6) | 158 (8.8) | 158 (8.8) | 108 (6.1) | 108 (6.1) | 261 (14.8) | 503 (28.6) | 549 (31.4) | 274 (15.8) |
| 25 to <30 (overweight) | 3,244 (8.2) | 29 (0.9) | 254 (7.7) | 188 (5.7) | 82 (2.5) | 201 (6.1) | 175 (5.4) | 125 (3.9) | 160 (4.9) | 368 (11.4) | 636 (19.8) | 718 (22.4) | 308 (9.7) |
| 30 to <35 (obese I) | 2,262 (7.7) | 13 (0.5) | 140 (5.8) | 127 (5.2) | 64 (2.7) | 146 (6.1) | 125 (5.2) | 81 (3.4) | 117 (4.9) | 291 (12.2) | 466 (19.6) | 481 (20.4) | 211 (9.0) |
| 35 to <40 (obese II) | 1,091 (8.9) | 12 (1.2) | 78 (7.6) | 41 (4.0) | 30 (2.9) | 57 (5.6) | 92 (9.1) | 33 (3.3) | 54 (5.4) | 120 (12.0) | 236 (23.6) | 235 (23.6) | 103 (10.5) |
| ≥40 (obese III) | 669 (12.5) | 5 (1.1) | 53 (11.9) | 32 (7.2) | 20 (4.5) | 46 (10.4) | 39 (8.9) | 18 (4.1) | 49 (11.2) | 93 (21.4) | 132 (30.5) | 123 (28.6) | 59 (13.9) |
| CCI | | | | | | | | | | | | | |
| 0 to 1 | 1,450 (2.0) | 9 (0.1) | 94 (1.6) | 98 (1.6) | 43 (0.7) | 88 (1.5) | 93 (1.5) | 61 (1.0) | 73 (1.2) | 176 (3.0) | 279 (4.7) | 308 (5.2) | 128 (2.2) |
| 2 to 3 | 2,156 (11.8) | 20 (1.3) | 157 (10.4) | 115 (7.6) | 50 (3.3) | 122 (8.2) | 127 (8.5) | 78 (5.3) | 125 (8.5) | 244 (16.7) | 470 (32.2) | 429 (29.6) | 219 (15.3) |
| 4 to 5 | 2,331 (25.3) | 11 (1.4) | 161 (21.1) | 131 (17.3) | 75 (10.0) | 138 (18.4) | 137 (18.4) | 88 (11.9) | 89 (12.1) | 284 (39.0) | 467 (64.3) | 512 (71.2) | 238 (33.6) |
| ≥6 | 4,293 (46.2) | 43 (5.6) | 363 (47.5) | 287 (38.0) | 127 (17.0) | 279 (37.7) | 247 (33.8) | 148 (20.5) | 213 (29.9) | 461 (65.3) | 803 (114.5) | 923 (133.3) | 399 (58.9) |

* VA enrollees at risk are those who are still alive at the beginning of each time period and had not been infected with SARS-CoV-2 more than 30 days before the beginning of the time period.

† Categorized according to the 10 Federal Regions drawn up by the Federal Emergency Management Agency: 1 (CT, MA, ME, NH, RI, and VT), 2 (NJ, NY, and PR), 3 (DC, DE, MD, PA, VA, and WV), 4 (AL, FL, GA, KY, MS, NC, SC, and TN), 5 (IL, IN, MI, MN, OH, and WI), 6 (AR, LA, NM, OK, and TX), 7 (IA, KS, MO, and NE), 8 (CO, MT, ND, SD, UT, and WY), 9 (AZ, CA, GU, HI, and NV), and 10 (AK, ID, OR, and WA).

AI/AN, American Indian/Alaska Native; CCI, Charlson comorbidity index; PI/NH, Pacific Islander/Native Hawaiian; SARS-CoV-2, Severe Acute Respiratory Syndrome Coronavirus 2; VA, Veterans Affairs.

attenuated over time and actually reversed after September 2020. However, we also show that during the third wave of the pandemic, from December 2020 to March 2021, Black race was again associated with higher risk of SARS-CoV-2 infection and mortality, albeit at a much lower level than in the early period of the pandemic.

It is unclear why such a dramatic reduction in risk of infection in Black persons relative to White persons occurred in the space of 9 months between February and November 2020 [25,26]. Although the pandemic shifted from early urban epicenters with high percentages of Black persons to a broad, nationwide distribution in both urban and rural communities with lower percentages of Black persons, our results persisted even after adjustment for both urban/rural location and geographic region. Some factors that mediate racial inequities and contributed to increased risk in Black persons are unlikely to be reversed quickly, such as occupation (e.g., disproportionate representation in essential work settings with high exposure risk such as healthcare facilities, farms, factories, grocery stores, and public transportation), inability to work from home, housing (e.g., crowded housing, multigenerational households, and residence in densely populated neighborhoods with high infection rates), and reliance on public transportation. Factors that can change quickly include those related to prophylactic measures (masking, handwashing, physical distancing, and adhering to stay-at-home orders), disease awareness and behavior (e.g., limiting unnecessary exposure to crowds), and increased rates of testing (leading to early identification and reduction in transmission rates [27]).

Our comparisons of Black versus White VA enrollees need to be interpreted with caution when trying to extrapolate to differences by race in the US population as a whole. Black Veterans report higher median incomes (US$44,000 versus US$26,000), lower unemployment (3% versus 5%), and lower poverty levels (10% versus 21%) than non-Veteran Black adults[28].

**Table 4. Trends over time in the associations (AORs\*) of sociodemographic characteristics and comorbidity burden with risk of SARS-CoV-2–related mortality in a cohort of 9.1 million VA enrollees from February 2020 to February 2021.**

| | AORs* for SARS-CoV-2–related mortality (95% CI) | | | | | | |
|---|---|---|---|---|---|---|---|
| | Entire period: February 2020 to February 2021 N = 9,127,673 | February to April 2020 N = 9,127,673 | May to June 2020 N = 9,059,880 | July to August 2020 N = 8,998,307 | September to October 2020 N = 8,922,764 | November to December 2020 N = 8,857,881 | January to February 2021 N = 8,790,422 |
| **Sex** | | | | | | | |
| Female | 1 | 1 | 1 | 1 | 1 | 1 | 1 |
| Male | 1.59 (1.38 to 1.82) | 2.14 (1.28 to 3.59) | 1.21 (0.81 to 1.81) | 1.52 (1.05 to 2.21) | 1.96 (1.15 to 3.35) | 1.50 (1.17 to 1.94) | 1.68 (1.30 to 2.16) |
| **Age (years)** | | | | | | | |
| 25 | 0.09 (0.04 to 0.17) | 0.02 (0.00 to 0.63) | 0.04 (0.00 to 0.82) | 0.01 (0.00 to 0.18) | 0.01 (0.00 to 0.45) | 0.03 (0.01 to 0.16) | 0.32 (0.14 to 0.73) |
| 35 | 0.31 (0.23 to 0.42) | 0.15 (0.03 to 0.73) | 0.22 (0.06 to 0.82) | 0.14 (0.04 to 0.43) | 0.10 (0.02 to 0.64) | 0.20 (0.10 to 0.41) | 0.56 (0.39 to 0.80) |
| 45 (reference) | 1 | 1 | 1 | 1 | 1 | 1 | 1 |
| 55 | 2.39 (2.14 to 2.67) | 3.47 (1.97 to 6.10) | 3.04 (1.88 to 4.92) | 3.12 (2.09 to 4.67) | 3.63 (1.86 to 7.08) | 2.69 (2.09 to 3.46) | 1.88 (1.61 to 2.19) |
| 65 | 5.05 (4.27 to 5.98) | 6.87 (3.55 to 13.30) | 6.72 (3.53 to 12.80) | 5.83 (3.57 to 9.52) | 7.18 (3.42 to 15.08) | 5.58 (3.98 to 7.82) | 4.06 (3.05 to 5.41) |
| 75 | 11.40 (9.77 to 13.30) | 11.68 (6.23 to 21.91) | 14.25 (7.81 to 25.99) | 12.30 (7.77 to 19.48) | 16.86 (8.29 to 34.28) | 14.26 (10.41 to 19.52) | 9.51 (7.36 to 12.30) |
| 85 | 20.37 (17.48 to 23.74) | 20.46 (10.92 to 38.34) | 30.53 (16.82 to 55.41) | 22.64 (14.30 to 35.85) | 33.38 (16.43 to 67.82) | 27.31 (19.97 to 37.34) | 16.74 (12.97 to 21.61) |
| **Race** | | | | | | | |
| White | 1 | 1 | 1 | 1 | 1 | 1 | 1 |
| Black | 1.55 (1.47 to 1.64) | 3.85 (3.30 to 4.50) | 2.21 (1.89 to 2.60) | 1.84 (1.59 to 2.13) | 1.16 (0.95 to 1.43) | 1.08 (0.96 to 1.20) | 1.28 (1.16 to 1.42) |
| Asian | 0.82 (0.64 to 1.05) | 0.90 (0.29 to 2.83) | 1.39 (0.68 to 2.82) | 1.07 (0.59 to 1.96) | 1.40 (0.62 to 3.17) | 0.71 (0.44 to 1.16) | 0.60 (0.38 to 0.93) |
| AI/AN | 1.66 (1.39 to 1.98) | 1.93 (0.86 to 4.32) | 1.93 (1.00 to 3.73) | 1.67 (1.03 to 2.71) | 2.25 (1.37 to 3.71) | 1.57 (1.15 to 2.13) | 1.44 (1.04 to 2.01) |
| PI/NH | 1.03 (0.83 to 1.28) | 1.16 (0.48 to 2.80) | 0.97 (0.43 to 2.17) | 0.87 (0.46 to 1.62) | 0.76 (0.32 to 1.84) | 1.20 (0.85 to 1.71) | 0.96 (0.67 to 1.39) |
| Missing/unknown/refused | 0.85 (0.78 to 0.93) | 1.17 (0.82 to 1.66) | 0.77 (0.55 to 1.09) | 0.91 (0.71 to 1.17) | 0.73 (0.52 to 1.02) | 0.95 (0.81 to 1.11) | 0.71 (0.60 to 0.84) |
| **Ethnicity** | | | | | | | |
| Non-Hispanic | 1 | 1 | 1 | 1 | 1 | 1 | 1 |
| Hispanic | 1.57 (1.45 to 1.70) | 0.74 (0.55 to 0.99) | 1.20 (0.91 to 1.59) | 2.65 (2.19 to 3.20) | 1.42 (1.04 to 1.95) | 1.57 (1.35 to 1.82) | 1.60 (1.39 to 1.84) |
| Missing/unknown/refused | 0.46 (0.40 to 0.52) | 0.26 (0.15 to 0.44) | 0.40 (0.26 to 0.63) | 0.44 (0.31 to 0.63) | 0.45 (0.29 to 0.71) | 0.35 (0.28 to 0.44) | 0.65 (0.52 to 0.80) |
| **US Federal Region[†]** | | | | | | | |
| 1 | 0.90 (0.81 to 1.00) | 7.21 (5.37 to 9.69) | 2.61 (2.04 to 3.34) | 0.15 (0.08 to 0.26) | 0.12 (0.05 to 0.26) | 0.86 (0.69 to 1.07) | 0.62 (0.51 to 0.76) |
| 2 | 0.74 (0.68 to 0.81) | 7.93 (6.16 to 10.21) | 1.54 (1.21 to 1.94) | 0.08 (0.05 to 0.13) | 0.15 (0.09 to 0.25) | 0.51 (0.42 to 0.63) | 0.51 (0.43 to 0.60) |
| 3 | 0.89 (0.83 to 0.96) | 2.26 (1.68 to 3.05) | 1.24 (0.98 to 1.57) | 0.32 (0.24 to 0.42) | 0.59 (0.45 to 0.77) | 1.18 (1.02 to 1.37) | 0.86 (0.76 to 0.98) |
| 4 (reference) | 1 | 1 | 1 | 1 | 1 | 1 | 1 |
| 5 | 1.04 (0.98 to 1.11) | 2.92 (2.23 to 3.82) | 1.27 (1.03 to 1.57) | 0.33 (0.26 to 0.41) | 0.67 (0.54 to 0.84) | 2.00 (1.78 to 2.23) | 0.66 (0.58 to 0.75) |
| 6 | 1.31 (1.23 to 1.40) | 2.47 (1.84 to 3.31) | 0.89 (0.68 to 1.15) | 1.32 (1.14 to 1.53) | 1.09 (0.89 to 1.34) | 1.75 (1.54 to 1.98) | 1.04 (0.93 to 1.17) |
| 7 | 1.38 (1.27 to 1.51) | 1.23 (0.74 to 2.05) | 0.99 (0.69 to 1.41) | 0.51 (0.38 to 0.69) | 1.96 (1.58 to 2.44) | 2.48 (2.16 to 2.86) | 0.94 (0.80 to 1.11) |
| 8 | 1.16 (1.04 to 1.29) | 1.77 (0.98 to 3.18) | 1.60 (1.11 to 2.30) | 0.25 (0.15 to 0.41) | 1.62 (1.23 to 2.12) | 2.19 (1.85 to 2.59) | 0.57 (0.44 to 0.72) |
| 9 | 1.41 (1.32 to 1.51) | 1.16 (0.78 to 1.72) | 0.89 (0.68 to 1.18) | 1.05 (0.88 to 1.24) | 0.55 (0.41 to 0.74) | 1.79 (1.57 to 2.05) | 1.70 (1.52 to 1.89) |
| 10 | 0.76 (0.67 to 0.87) | 1.38 (0.73 to 2.60) | 0.81 (0.50 to 1.32) | 0.57 (0.40 to 0.82) | 0.69 (0.46 to 1.03) | 1.10 (0.88 to 1.37) | 0.57 (0.45 to 0.73) |
| **Urban versus rural** | | | | | | | |
| Rural | 1 | 1 | 1 | 1 | 1 | 1 | 1 |
| Urban | 1.00 (0.96 to 1.04) | 2.48 (2.08 to 2.96) | 1.81 (1.57 to 2.10) | 1.11 (0.99 to 1.24) | 0.72 (0.62 to 0.83) | 0.80 (0.74 to 0.86) | 0.95 (0.88 to 1.02) |
| **BMI (kg/m²)** | | | | | | | |
| 20 | 1.62 (1.53 to 1.71) | 1.53 (1.27 to 1.84) | 1.98 (1.68 to 2.33) | 1.68 (1.42 to 1.98) | 1.58 (1.29 to 1.95) | 1.77 (1.59 to 1.97) | 1.75 (1.58 to 1.94) |
| 25 (reference) | 1 | 1 | 1 | 1 | 1 | 1 | 1 |
| 30 | 1.00 (0.94 to 1.06) | 0.95 (0.78 to 1.17) | 0.92 (0.76 to 1.13) | 0.94 (0.79 to 1.13) | 0.94 (0.76 to 1.15) | 1.11 (0.99 to 1.23) | 0.95 (0.85 to 1.06) |
| 35 | 1.20 (1.13 to 1.27) | 1.00 (0.82 to 1.22) | 1.02 (0.84 to 1.24) | 1.28 (1.09 to 1.50) | 1.15 (0.95 to 1.39) | 1.32 (1.20 to 1.46) | 1.12 (1.02 to 1.24) |

(*Continued*)

**Table 4.** (Continued)

| | AORs* for SARS-CoV-2–related mortality (95% CI) | | | | | | |
|---|---|---|---|---|---|---|---|
| | Entire period: February 2020 to February 2021 N = 9,127,673 | February to April 2020 N = 9,127,673 | May to June 2020 N = 9,059,880 | July to August 2020 N = 8,998,307 | September to October 2020 N = 8,922,764 | November to December 2020 N = 8,857,881 | January to February 2021 N = 8,790,422 |
| 40 | 1.48 (1.40 to 1.57) | 1.35 (1.11 to 1.64) | 1.30 (1.07 to 1.59) | 1.58 (1.34 to 1.86) | 1.48 (1.22 to 1.80) | 1.64 (1.48 to 1.81) | 1.33 (1.20 to 1.47) |
| **CCI** | | | | | | | |
| 0 to 1 | 1 | 1 | 1 | 1 | 1 | 1 | 1 |
| 2 to 3 | 3.06 (2.86 to 3.27) | 3.32 (2.60 to 4.26) | 2.29 (1.82 to 2.88) | 2.85 (2.35 to 3.47) | 3.02 (2.42 to 3.77) | 3.21 (2.85 to 3.61) | 3.28 (2.90 to 3.72) |
| 4 to 5 | 5.58 (5.22 to 5.97) | 5.30 (4.12 to 6.81) | 4.79 (3.85 to 5.97) | 5.30 (4.36 to 6.42) | 4.47 (3.55 to 5.63) | 5.80 (5.15 to 6.54) | 6.63 (5.87 to 7.50) |
| ≥6 | 9.46 (8.89 to 10.07) | 10.46 (8.34 to 13.12) | 8.64 (7.07 to 10.56) | 9.44 (7.91 to 11.27) | 8.90 (7.24 to 10.94) | 9.59 (8.58 to 10.72) | 11.41 (10.18 to 12.79) |

* Adjusted for sex, age (modeled as restricted cubic splines with 5 knots at ages 30, 49, 64, 73, and 88 years), race, ethnicity, geographical region, urban/rural location, BMI (modeled as restricted cubic splines with 5 knots at BMIs of 21.3, 26.0, 29.0, 32.5, and 39.9 kg/m$^2$), and CCI.

† Categorized according to the 10 Federal Regions drawn up by the Federal Emergency Management Agency: 1 (CT, MA, ME, NH, RI, and VT), 2 (NJ, NY, and PR), 3 (DC, DE, MD, PA, VA, and WV), 4 (AL, FL, GA, KY, MS, NC, SC, and TN), 5 (IL, IN, MI, MN, OH, and WI), 6 (AR, LA, NM, OK, and TX), 7 (IA, KS, MO, and NE), 8 (CO, MT, ND, SD, UT, and WY), 9 (AZ, CA, GU, HI, and NV), and 10 (AK, ID, OR, and WA).

AI/AN, American Indian/Alaska Native; AOR, adjusted odds ratio; BMI, body mass index; CCI, Charlson comorbidity index; PI/NH, Pacific Islander/Native Hawaiian; SARS-CoV-2, Severe Acute Respiratory Syndrome Coronavirus 2; VA, Veterans Affairs.

However, Black VA enrollees have lower socioeconomic status than White VA enrollees including much higher unemployment (22.5% versus 6.8%) [29]. Levels of perceived discrimination in healthcare have been shown to be equally high among Black Veteran and non-Veteran groups in the US, despite access to comprehensive healthcare through the VA [30]. Furthermore, racial discrimination experienced during military service has been shown to be common among Black Veterans and associated with long-term impacts on self-reported physical health [31,32].

AI/AN persons had significant higher risk of infection than White persons early in the pandemic, but this association attenuated over time and actually reversed during the third wave of the pandemic between December 2020 and March 2021. However, SARS-CoV-2 mortality remained higher in AI/AN persons throughout the observation period.

Hispanic ethnicity was associated with increased risk of SARS-CoV-2 infection and mortality during almost every time period that we investigated. In contrast to Black race, the associations of Hispanic ethnicity with infection or mortality did not attenuate over time. This suggests that the factors listed above that mediate inequities in SARS-CoV-2 infection and mortality were not improved over time in Hispanic communities.

Big cities and metropolitan areas constituted the initial epicenters of SARS-CoV-2 infection in the US. However, the virus quickly spread throughout the US to both rural and urban areas, likely explaining the observed attenuation in the association between urban residence and SARS-CoV-2 infection and mortality over time. Unique challenges exist in both urban (dense population, housing distress, overcrowding, and public transport) and rural (geographic inaccessibility to SARS-CoV-2–related screening and care, higher disability, lack of social capital, and high-risk occupations such as meat and poultry processing [33]) that tend to raise risk of SARS-CoV-2 infection and mortality.

Comorbidity burden, estimated by the CCI, was one of the strongest risk factors for both SARS-CoV-2 infection and mortality. Furthermore, after May 2020, we did not observe any attenuation over time in the associations of high CCI with SARS-CoV-2–related mortality. It is possible that increased likelihood of testing persons with high comorbidity burden even if minimally symptomatic or asymptomatic may have contributed to the observed association with risk of infection. However, this does not explain the strong associations between high

**Table 5. SARS-CoV-2 case fatality presented for VA enrollees who tested positive for SARS-CoV-2 each monthly period from February 2020 to February 2021.**

| Time period and number of VA enrollees with positive SARS-CoV-2 test | Cohort characteristics N = 206,789 | Number of SARS-CoV-2–related deaths among persons testing positive each month (and mortality per 100 persons) | | | | | | | | | | | | |
|---|---|---|---|---|---|---|---|---|---|---|---|---|---|---|
| | | Entire period: February 2020 to February 2021 N = 206,789 | February to March 2020 N = 2,470 | April 2020 N = 7,346 | May 2020 N = 4,957 | June 2020 N = 7,634 | July 2020 N = 16,105 | August 2020 N = 9,189 | September 2020 N = 7,526 | October 2020 N = 13,523 | November 2020 N = 33,553 | December 2020 N = 47,357 | January 2021 N = 39,431 | February 2021 N = 17,698 |
| SARS-CoV-2–related deaths[*], n (%) | | 10,429 (5.0%) | 312 (12.6) | 875 (11.9) | 442 (8.9) | 376 (4.9) | 724 (4.5) | 459 (5.0) | 379 (5.0) | 695 (5.1) | 1,491 (4.4) | 2,398 (5.1) | 1,732 (4.4) | 546 (3.1) |
| **Sex** | | | | | | | | | | | | | | |
| Female | 21,033 (10.2) | 211 (1.0) | 7 (3.2) | 16 (2.3) | 14 (3.1) | 8 (0.9) | 18 (0.9) | 9 (0.9) | 4 (0.6) | 16 (1.3) | 22 (0.7) | 48 (1.0) | 35 (0.9) | 14 (0.8) |
| Male | 185,756 (89.8) | 10,218 (5.5) | 305 (13.5) | 859 (12.9) | 428 (9.5) | 368 (5.4) | 706 (5.0) | 450 (5.5) | 375 (5.5) | 679 (5.5) | 1,469 (4.9) | 2,350 (5.5) | 1,697 (4.8) | 532 (3.4) |
| **Age (years)** | | | | | | | | | | | | | | |
| 18 to 24 | 1,293 (0.6) | 0 (0.0) | 0 (0.0) | 0 (0.0) | 0 (0.0) | 0 (0.0) | 0 (0.0) | 0 (0.0) | 0 (0.0) | 0 (0.0) | 0 (0.0) | 0 (0.0) | 0 (0.0) | 0 (0.0) |
| 25 to 34 | 15,919 (7.7) | 11 (0.1) | 0 (0.0) | 0 (0.0) | 0 (0.0) | 0 (0.0) | 0 (0.0) | 0 (0.0) | 0 (0.0) | 0 (0.0) | 1 (0.0) | 4 (0.1) | 1 (0.0) | 5 (0.4) |
| 35 to 44 | 23,339 (11.3) | 40 (0.2) | 0 (0.0) | 1 (0.2) | 3 (0.7) | 2 (0.2) | 3 (0.1) | 0 (0.0) | 1 (0.1) | 1 (0.1) | 2 (0.1) | 12 (0.2) | 11 (0.3) | 4 (0.2) |
| 45 to 54 | 29,783 (14.4) | 163 (0.5) | 11 (3.0) | 13 (1.5) | 7 (1.3) | 5 (0.4) | 15 (0.6) | 8 (0.6) | 9 (0.8) | 6 (0.3) | 17 (0.3) | 34 (0.5) | 24 (0.4) | 14 (0.6) |
| 55 to 64 | 39,340 (19.0) | 762 (1.9) | 46 (8.2) | 56 (3.7) | 29 (2.7) | 35 (2.5) | 68 (2.2) | 39 (2.2) | 22 (1.6) | 47 (1.9) | 79 (1.3) | 161 (1.8) | 135 (1.8) | 45 (1.3) |
| 65 to 74 | 57,869 (28.0) | 3,432 (5.9) | 124 (18.0) | 274 (13.0) | 122 (8.7) | 123 (6.8) | 257 (6.4) | 146 (5.7) | 125 (5.6) | 221 (5.6) | 504 (5.3) | 791 (6.0) | 562 (5.0) | 183 (3.5) |
| 75 to 84 | 26,793 (13.0) | 3,045 (11.4) | 66 (23.7) | 225 (22.0) | 102 (14.7) | 104 (13.1) | 189 (12.1) | 135 (12.0) | 106 (10.6) | 218 (12.1) | 465 (10.6) | 735 (11.5) | 540 (10.0) | 160 (6.7) |
| ≥85 | 12,453 (6.0) | 2,976 (23.9) | 65 (46.1) | 306 (37.7) | 179 (31.7) | 107 (25.2) | 192 (25.4) | 131 (25.3) | 116 (25.8) | 202 (25.9) | 423 (22.3) | 661 (23.5) | 459 (20.2) | 135 (13.1) |
| **Race** | | | | | | | | | | | | | | |
| White | 138,364 (66.9) | 7,416 (5.4) | 120 (12.6) | 522 (13.8) | 296 (10.8) | 244 (5.5) | 480 (5.2) | 307 (5.4) | 290 (5.6) | 547 (5.5) | 1,156 (4.7) | 1,778 (5.4) | 1,282 (4.8) | 394 (3.3) |
| Black | 46,288 (22.4) | 1,975 (4.3) | 170 (13.1) | 290 (9.9) | 110 (6.4) | 92 (4.0) | 166 (3.4) | 95 (3.7) | 55 (3.4) | 77 (3.4) | 162 (3.0) | 378 (4.2) | 280 (3.3) | 100 (2.6) |
| Asian | 2,069 (1.0) | 66 (3.2) | 2 (7.4) | 4 (7.3) | 4 (10.8) | 1 (1.3) | 8 (4.8) | 6 (7.4) | 3 (4.3) | 1 (1.1) | 5 (1.8) | 20 (3.6) | 10 (2.3) | 2 (1.0) |
| AI/AN | 1,955 (0.9) | 125 (6.4) | 2 (14.3) | 7 (13.0) | 3 (6.3) | 7 (7.6) | 8 (5.2) | 9 (11.4) | 7 (9.2) | 10 (6.8) | 16 (4.7) | 36 (8.2) | 14 (4.0) | 6 (3.7) |
| PI/NH | 1,948 (0.9) | 90 (4.6) | 2 (15.4) | 5 (8.8) | 2 (4.5) | 3 (3.9) | 6 (3.8) | 5 (5.0) | 0 (0.0) | 6 (5.2) | 17 (5.6) | 23 (5.2) | 14 (3.6) | 7 (4.0) |
| Missing/unknown/refused | 16,165 (7.8) | 757 (4.7) | 16 (9.6) | 47 (9.7) | 27 (7.5) | 29 (4.4) | 56 (4.0) | 37 (5.3) | 24 (4.3) | 54 (5.4) | 135 (5.3) | 163 (4.2) | 132 (4.2) | 37 (2.8) |
| **Ethnicity** | | | | | | | | | | | | | | |
| Non-Hispanic | 177,958 (86.1) | 9,274 (5.2) | 284 (13.4) | 806 (12.5) | 401 (9.2) | 310 (5.1) | 607 (4.7) | 411 (5.3) | 349 (5.3) | 628 (5.3) | 1,351 (4.6) | 2,124 (5.2) | 1,513 (4.5) | 490 (3.2) |
| Hispanic | 20,751 (10.0) | 759 (3.7) | 24 (8.8) | 43 (6.6) | 26 (6.1) | 51 (4.1) | 91 (3.5) | 30 (2.9) | 18 (2.8) | 45 (3.9) | 91 (3.2) | 174 (3.7) | 131 (3.5) | 35 (2.4) |
| Missing/unknown/refused | 8,080 (3.9) | 396 (4.9) | 4 (5.1) | 26 (10.2) | 15 (9.0) | 15 (5.0) | 26 (4.4) | 18 (5.5) | 12 (4.5) | 22 (4.4) | 49 (3.9) | 100 (5.1) | 88 (5.3) | 21 (2.9) |
| **US Federal Region[†]** | | | | | | | | | | | | | | |
| 1 | 7,004 (3.4) | 427 (6.1) | 10 (10.5) | 150 (17.6) | 34 (8.5) | 8 (4.1) | 9 (6.3) | 2 (2.0) | 4 (4.0) | 5 (2.3) | 41 (4.3) | 77 (4.6) | 68 (4.5) | 19 (2.5) |
| 2 | 10,959 (5.3) | 718 (6.6) | 112 (18.2) | 217 (12.6) | 55 (9.3) | 14 (5.2) | 7 (2.1) | 6 (2.8) | 6 (2.9) | 16 (4.2) | 39 (3.1) | 119 (5.5) | 91 (4.3) | 36 (3.3) |
| 3 | 17,339 (8.4) | 931 (5.4) | 25 (13.5) | 96 (11.5) | 55 (8.4) | 34 (7.4) | 32 (4.1) | 21 (3.6) | 30 (6.3) | 49 (5.9) | 123 (4.9) | 243 (5.4) | 168 (4.5) | 55 (3.0) |
| 4 | 53,231 (25.7) | 2,447 (4.6) | 26 (7.2) | 110 (10.1) | 100 (9.9) | 94 (3.8) | 270 (4.5) | 191 (5.6) | 123 (5.3) | 146 (4.8) | 213 (3.8) | 537 (4.8) | 485 (4.2) | 152 (2.8) |
| 5 | 32,309 (15.6) | 1,592 (4.9) | 60 (14.5) | 146 (11.9) | 83 (8.7) | 26 (4.3) | 52 (4.3) | 49 (4.9) | 40 (3.7) | 149 (4.9) | 389 (4.6) | 336 (4.5) | 188 (4.1) | 74 (3.4) |
| 6 | 31,762 (15.4) | 1,594 (5.0) | 50 (10.9) | 65 (9.9) | 39 (8.0) | 82 (4.7) | 192 (4.8) | 89 (5.3) | 50 (4.1) | 135 (6.2) | 234 (5.3) | 324 (5.0) | 254 (4.1) | 80 (3.4) |
| 7 | 13,984 (6.8) | 718 (5.1) | 7 (12.1) | 23 (9.7) | 22 (8.8) | 10 (4.5) | 20 (3.3) | 32 (4.7) | 66 (7.7) | 80 (5.6) | 159 (4.4) | 176 (5.8) | 96 (4.8) | 27 (2.9) |
| 8 | 8,623 (4.2) | 394 (4.6) | 4 (5.4) | 18 (8.7) | 26 (14.5) | 2 (1.4) | 10 (3.6) | 14 (5.8) | 23 (5.6) | 57 (5.2) | 109 (4.2) | 79 (4.1) | 40 (4.0) | 12 (2.3) |
| 9 | 26,230 (12.7) | 1,358 (5.2) | 13 (7.9) | 35 (9.0) | 23 (6.7) | 99 (7.2) | 110 (4.7) | 43 (4.4) | 24 (3.8) | 37 (4.3) | 141 (4.5) | 446 (5.8) | 308 (5.1) | 79 (3.5) |
| 10 | 5,348 (2.6) | 250 (4.7) | 5 (11.4) | 15 (11.3) | 5 (5.4) | 7 (4.8) | 22 (5.7) | 12 (4.1) | 13 (5.9) | 21 (4.8) | 43 (3.9) | 61 (4.8) | 34 (4.4) | 12 (2.6) |
| **Urban versus rural** | | | | | | | | | | | | | | |
| Rural | 99,646 (48.2) | 5,106 (5.1) | 57 (9.8) | 196 (10.5) | 137 (8.4) | 156 (5.3) | 335 (4.9) | 261 (5.6) | 233 (5.6) | 425 (5.7) | 870 (5.0) | 1,289 (5.4) | 867 (4.5) | 280 (3.2) |
| Urban | 107,143 (51.8) | 5,323 (5.0) | 255 (13.5) | 679 (12.4) | 305 (9.1) | 220 (4.7) | 389 (4.2) | 198 (4.4) | 146 (4.3) | 270 (4.5) | 621 (3.9) | 1,109 (4.7) | 865 (4.3) | 266 (3.0) |
| **BMI (kg/m²)** | | | | | | | | | | | | | | |

*(Continued)*

**Table 5.** (Continued)

| Time period and number of VA enrollees with positive SARS-CoV-2 test | Cohort characteristics N = 206,789 | Number of SARS-CoV-2–related deaths among persons testing positive each month (and mortality per 100 persons) | | | | | | | | | | | | |
|---|---|---|---|---|---|---|---|---|---|---|---|---|---|---|
| | | Entire period: February 2020 to February 2021 N = 206,789 | February to March 2020 N = 2,470 | April 2020 N = 7,346 | May 2020 N = 4,957 | June 2020 N = 7,634 | July 2020 N = 16,105 | August 2020 N = 9,189 | September 2020 N = 7,526 | October 2020 N = 13,523 | November 2020 N = 33,553 | December 2020 N = 47,357 | January 2021 N = 39,431 | February 2021 N = 17,698 |
| <18.5 (underweight) | 1,897 (0.9) | 312 (16.4) | 13 (34.2) | 31 (20.9) | 23 (22.8) | 20 (21.1) | 12 (10.4) | 13 (13.3) | 13 (17.6) | 15 (14.3) | 39 (17.5) | 57 (15.2) | 57 (15.5) | 19 (11.9) |
| 18.5 to <25 (normal weight) | 31,118 (15.0) | 2,704 (8.7) | 78 (18.9) | 269 (18.6) | 152 (14.8) | 104 (8.5) | 173 (7.2) | 125 (9.2) | 108 (9.2) | 143 (7.7) | 344 (7.4) | 617 (8.9) | 452 (7.7) | 139 (5.1) |
| 25 to <30 (overweight) | 67,238 (32.5) | 3,296 (4.9) | 93 (11.7) | 288 (12.3) | 129 (8.1) | 111 (4.4) | 235 (4.5) | 135 (4.6) | 121 (4.9) | 216 (4.9) | 465 (4.3) | 807 (5.2) | 526 (4.1) | 170 (3.0) |
| 30 to <35 (obese I) | 59,700 (28.9) | 2,312 (3.9) | 67 (9.9) | 164 (8.6) | 78 (6.2) | 85 (3.9) | 156 (3.3) | 97 (3.6) | 85 (3.9) | 184 (4.6) | 358 (3.6) | 518 (3.8) | 405 (3.5) | 115 (2.3) |
| 35 to <40 (obese II) | 29,939 (14.5) | 1,118 (3.7) | 39 (11.0) | 69 (7.3) | 39 (6.2) | 30 (2.9) | 90 (3.8) | 69 (4.9) | 28 (2.7) | 75 (3.7) | 168 (3.4) | 250 (3.6) | 198 (3.5) | 63 (2.5) |
| ≥40 (obese III) | 16,897 (8.2) | 687 (4.1) | 22 (11.4) | 54 (9.7) | 21 (6.3) | 26 (4.4) | 58 (4.3) | 20 (2.7) | 24 (3.8) | 62 (5.3) | 117 (4.1) | 149 (3.8) | 94 (3.0) | 40 (2.8) |
| **CCI** | | | | | | | | | | | | | | |
| 0 to 1 | 93,603 (45.3) | 1,480 (1.6) | 38 (4.0) | 108 (4.1) | 78 (4.3) | 59 (1.6) | 102 (1.3) | 61 (1.5) | 67 (2.0) | 102 (1.7) | 217 (1.4) | 351 (1.6) | 217 (1.2) | 80 (1.0) |
| 2 to 3 | 49,109 (23.7) | 2,208 (4.5) | 71 (12.9) | 160 (9.7) | 87 (7.5) | 63 (4.0) | 149 (4.0) | 98 (4.7) | 87 (4.8) | 156 (4.8) | 316 (4.0) | 521 (4.6) | 364 (3.7) | 136 (3.1) |
| 4 to 5 | 28,738 (13.9) | 2,369 (8.2) | 55 (15.9) | 188 (16.3) | 90 (11.2) | 90 (9.1) | 171 (8.6) | 110 (8.5) | 72 (6.7) | 135 (7.4) | 369 (8.0) | 540 (8.2) | 440 (7.8) | 109 (4.5) |
| ≥6 | 35,339 (17.1) | 4,372 (12.4) | 148 (24.0) | 419 (21.8) | 187 (15.7) | 164 (12.9) | 302 (12.1) | 190 (11.9) | 153 (11.5) | 302 (13.0) | 589 (11.1) | 986 (12.5) | 711 (10.9) | 221 (7.7) |

* The number of deaths each month are different than those shown in Table 3, because Table 5 shows the number of persons who tested positive each month (same as in Table 1) and among them those who died within 30 days—some of these deaths occurred in the following month, but patients are grouped based on the date of infection.

† Categorized according to the 10 Federal Regions drawn up by the Federal Emergency Management Agency: 1 (CT, MA, ME, NH, RI, and VT), 2 (NJ, NY, and PR), 3 (DC, DE, MD, PA, VA, and WV), 4 (AL, FL, GA, KY, MS, NC, SC, and TN), 5 (IL, IN, MI, MN, OH, and WI), 6 (AR, LA, NM, OK, and TX), 7 (IA, KS, MO, and NE), 8 (CO, MT, ND, SD, UT, and WY), 9 (AZ, CA, GU, HI, and NV), and 10 (AK, ID, OR, and WA).

AI/AN, American Indian/Alaska Native; CCI, Charlson comorbidity index; PI/NH, Pacific Islander/Native Hawaiian; SARS-CoV-2, Severe Acute Respiratory Syndrome Coronavirus 2; VA, Veterans Affairs.

CCI and SARS-CoV-2–related mortality and case fatality that we observed. Most of the component comorbidities that make up the CCI have been individually associated with adverse COVID-19 outcomes, especially diabetes, congestive heart failure, chronic pulmonary disease, cerebrovascular disease, liver disease, kidney disease, and malignancy [1,34–39]. The cumulative burden of these comorbidities appears to have a dramatic effect on SARS-CoV-2–related mortality. The associations between high CCI and risk of infection appear to decline steadily from January to March 2021. This may be related to initiation of vaccination that was restricted to high-risk groups in that time period.

Compared to age 45 years (the reference age in our analysis), both older and younger persons had significantly lower risk of testing positive in all time periods after adjusting for sex, race, ethnicity, geographical region, urban/rural location, BMI, and CCI. It is interesting that risk of testing positive in older persons declined even further in March 2021, which may be related to initiation of vaccination in older age groups during that time period. Similar associations between age ≥65 and lower risk of infection have been described with influenza virus [40] and are likely related to lower risk of exposure to SARS-CoV-2 in older persons. Despite the lower incidence of infection, older age was the strongest risk factor for SARS-CoV-2–related mortality (driven by much higher case fatality in older persons), an association that did not change in magnitude over time.

Our results should be interpreted in the context of some important limitations. It is impossible to identify all cases of SARS-CoV-2 infection in a population, since many are asymptomatic and even some symptomatic cases do not get confirmed by testing. We identified only the cases who were tested and identified as positive within the VA system (positive tests performed

**Table 6. Trends over time in the associations (AORs[*]) of sociodemographic characteristics and comorbidity burden with risk of SARS-CoV-2 case fatality among 206,789 VA enrollees who tested positive for SARS-CoV-2 from February 1, 2020 to February 28, 2021, presented overall and divided into time periods.**

| | AORs[*] for SARS-CoV-2–related death among persons testing positive (95% CI) | | | | | | |
|---|---|---|---|---|---|---|---|
| | Entire period: February 2020 to February 2021 $N = 206,789$ | February to April 2020 $N = 9,816$ | May to June 2020 $N = 12,591$ | July to August 2020 $N = 25,294$ | September to October 2020 $N = 21,049$ | November to December 2020 $N = 80,910$ | January to February 2021 $N = 57,129$ |
| **Sex** | | | | | | | |
| Female | 1 | 1 | 1 | 1 | 1 | 1 | 1 |
| Male | 1.73 (1.50 to 2.00) | 1.90 (1.25 to 2.90) | 1.36 (0.88 to 2.09) | 1.62 (1.10 to 2.39) | 1.67 (1.07 to 2.62) | 1.72 (1.35 to 2.20) | 1.63 (1.22 to 2.18) |
| **Age (years)** | | | | | | | |
| 25 | 0.15 (0.08 to 0.29) | 0.02 (0.00 to 0.47) | 0.05 (0.00 to 0.67) | 0.02 (0.00 to 0.34) | 0.01 (0.00 to 0.56) | 0.21 (0.08 to 0.57) | 0.36 (0.14 to 0.95) |
| 35 | 0.38 (0.29 to 0.50) | 0.15 (0.04 to 0.61) | 0.22 (0.07 to 0.72) | 0.16 (0.05 to 0.53) | 0.10 (0.02 to 0.66) | 0.44 (0.28 to 0.67) | 0.57 (0.38 to 0.86) |
| 45 (reference) | 1 | 1 | 1 | 1 | 1 | 1 | 1 |
| 55 | 3.07 (2.76 to 3.42) | 4.42 (2.65 to 7.38) | 3.62 (2.35 to 5.58) | 4.25 (2.74 to 6.61) | 5.18 (2.64 to 10.15) | 2.91 (2.44 to 3.46) | 2.36 (1.97 to 2.82) |
| 65 | 9.34 (7.87 to 11.09) | 12.05 (6.59 to 22.04) | 9.69 (5.29 to 17.77) | 11.83 (6.93 to 20.19) | 15.48 (7.41 to 32.35) | 9.44 (7.05 to 12.64) | 6.45 (4.65 to 8.95) |
| 75 | 22.58 (19.33 to 26.38) | 23.89 (13.42 to 42.53) | 20.84 (11.91 to 36.44) | 25.64 (15.46 to 42.52) | 37.64 (18.47 to 76.71) | 25.21 (19.33 to 32.88) | 14.75 (10.99 to 19.79) |
| 85 | 50.09 (42.87 to 58.52) | 46.48 (26.11 to 82.73) | 46.14 (26.45 to 80.47) | 50.60 (30.48 to 84.01) | 79.89 (39.17 to 162.93) | 50.33 (38.62 to 65.60) | 27.51 (20.51 to 36.91) |
| **Race** | | | | | | | |
| White | 1 | 1 | 1 | 1 | 1 | 1 | 1 |
| Black | 1.13 (1.06 to 1.19) | 2.56 (2.23 to 2.93) | 1.37 (1.15 to 1.64) | 1.27 (1.09 to 1.47) | 0.74 (0.61 to 0.90) | 0.86 (0.78 to 0.95) | 0.88 (0.78 to 0.99) |
| Asian | 1.50 (1.14 to 1.98) | 1.98 (0.86 to 4.56) | 1.09 (0.44 to 2.68) | 2.96 (1.70 to 5.15) | 1.31 (0.48 to 3.56) | 1.34 (0.88 to 2.03) | 0.90 (0.50 to 1.62) |
| AI/AN | 1.76 (1.44 to 2.14) | 2.04 (1.04 to 3.99) | 1.86 (0.99 to 3.52) | 1.83 (1.12 to 2.99) | 1.95 (1.20 to 3.19) | 1.71 (1.28 to 2.27) | 1.09 (0.70 to 1.71) |
| PI/NH | 1.19 (0.95 to 1.49) | 1.42 (0.67 to 3.03) | 0.81 (0.33 to 1.98) | 1.19 (0.65 to 2.17) | 0.79 (0.35 to 1.77) | 1.33 (0.96 to 1.83) | 1.11 (0.71 to 1.72) |
| Missing/unknown/refused | 1.16 (1.06 to 1.27) | 1.35 (1.01 to 1.80) | 1.05 (0.77 to 1.43) | 1.22 (0.95 to 1.55) | 1.19 (0.91 to 1.55) | 1.17 (1.01 to 1.34) | 0.99 (0.82 to 1.20) |
| **Ethnicity** | | | | | | | |
| Non-Hispanic | 1 | 1 | 1 | 1 | 1 | 1 | 1 |
| Hispanic | 1.21 (1.11 to 1.31) | 0.74 (0.57 to 0.96) | 1.48 (1.15 to 1.91) | 1.73 (1.41 to 2.13) | 1.15 (0.88 to 1.50) | 1.13 (0.99 to 1.30) | 1.11 (0.94 to 1.31) |
| Missing/unknown/refused | 1.00 (0.88 to 1.13) | 0.75 (0.50 to 1.11) | 1.05 (0.70 to 1.59) | 0.98 (0.70 to 1.38) | 0.79 (0.53 to 1.16) | 0.95 (0.78 to 1.15) | 1.36 (1.08 to 1.71) |
| **US Federal Region[†]** | | | | | | | |
| 1 | 0.99 (0.88 to 1.10) | 7.14 (5.61 to 9.08) | 1.09 (0.77 to 1.53) | 0.13 (0.07 to 0.24) | 0.19 (0.10 to 0.36) | 0.90 (0.74 to 1.10) | 0.76 (0.60 to 0.96) |
| 2 | 1.01 (0.92 to 1.11) | 7.69 (6.24 to 9.48) | 0.97 (0.73 to 1.29) | 0.09 (0.05 to 0.15) | 0.29 (0.19 to 0.46) | 0.76 (0.64 to 0.91) | 0.70 (0.57 to 0.85) |
| 3 | 1.00 (0.92 to 1.08) | 2.11 (1.65 to 2.71) | 1.14 (0.89 to 1.47) | 0.30 (0.22 to 0.40) | 0.78 (0.60 to 1.00) | 1.32 (1.16 to 1.50) | 0.92 (0.79 to 1.08) |
| 4 (reference) | 1 | 1 | 1 | 1 | 1 | 1 | 1 |
| 5 | 0.85 (0.79 to 0.91) | 2.09 (1.67 to 2.60) | 0.75 (0.59 to 0.95) | 0.29 (0.23 to 0.36) | 0.91 (0.75 to 1.09) | 1.28 (1.15 to 1.43) | 0.53 (0.46 to 0.62) |
| 6 | 1.07 (1.00 to 1.15) | 1.56 (1.21 to 2.00) | 1.02 (0.81 to 1.28) | 0.95 (0.82 to 1.11) | 1.10 (0.90 to 1.33) | 1.20 (1.07 to 1.35) | 0.85 (0.74 to 0.97) |
| 7 | 0.96 (0.88 to 1.05) | 0.91 (0.61 to 1.35) | 0.59 (0.40 to 0.86) | 0.38 (0.28 to 0.50) | 1.66 (1.35 to 2.04) | 1.41 (1.24 to 1.62) | 0.61 (0.50 to 0.75) |
| 8 | 0.92 (0.82 to 1.04) | 1.40 (0.89 to 2.22) | 0.92 (0.62 to 1.38) | 0.30 (0.20 to 0.46) | 1.54 (1.20 to 1.99) | 1.36 (1.15 to 1.60) | 0.45 (0.34 to 0.60) |
| 9 | 1.15 (1.07 to 1.24) | 0.73 (0.52 to 1.02) | 1.17 (0.92 to 1.48) | 0.61 (0.51 to 0.74) | 0.46 (0.35 to 0.62) | 1.65 (1.47 to 1.85) | 1.25 (1.09 to 1.43) |
| 10 | 0.96 (0.83 to 1.10) | 1.87 (1.16 to 3.01) | 0.63 (0.35 to 1.13) | 0.71 (0.50 to 1.01) | 1.09 (0.76 to 1.57) | 1.24 (1.00 to 1.53) | 0.64 (0.47 to 0.87) |
| **Urban versus rural** | | | | | | | |
| Rural | 1 | 1 | 1 | 1 | 1 | 1 | 1 |
| Urban | 1.03 (0.98 to 1.07) | 2.24 (1.93 to 2.60) | 1.59 (1.37 to 1.86) | 1.08 (0.96 to 1.22) | 0.81 (0.71 to 0.93) | 0.84 (0.78 to 0.90) | 0.98 (0.90 to 1.07) |
| **BMI (kg/m²)** | | | | | | | |
| 20 | 1.44 (1.35 to 1.53) | 1.30 (1.11 to 1.53) | 1.58 (1.33 to 1.87) | 1.20 (1.01 to 1.43) | 1.35 (1.12 to 1.62) | 1.29 (1.16 to 1.42) | 1.42 (1.26 to 1.60) |
| 25 (reference) | 1 | 1 | 1 | 1 | 1 | 1 | 1 |
| 30 | 0.96 (0.90 to 1.02) | 0.96 (0.80 to 1.15) | 0.84 (0.68 to 1.03) | 0.83 (0.69 to 0.99) | 1.14 (0.94 to 1.37) | 0.99 (0.90 to 1.09) | 0.92 (0.80 to 1.04) |
| 35 | 1.12 (1.06 to 1.19) | 0.94 (0.79 to 1.11) | 0.94 (0.76 to 1.16) | 1.17 (0.99 to 1.38) | 1.20 (1.01 to 1.44) | 1.13 (1.03 to 1.24) | 1.13 (1.00 to 1.27) |

(*Continued*)

**Table 6.** (Continued)

| | AORs[*] for SARS-CoV-2–related death among persons testing positive (95% CI) | | | | | | |
|---|---|---|---|---|---|---|---|
| | Entire period: February 2020 to February 2021 N = 206,789 | February to April 2020 N = 9,816 | May to June 2020 N = 12,591 | July to August 2020 N = 25,294 | September to October 2020 N = 21,049 | November to December 2020 N = 80,910 | January to February 2021 N = 57,129 |
| 40 | 1.35 (1.27 to 1.43) | 1.16 (0.98 to 1.39) | 1.13 (0.92 to 1.40) | 1.35 (1.14 to 1.60) | 1.48 (1.24 to 1.77) | 1.32 (1.20 to 1.45) | 1.29 (1.14 to 1.46) |
| **CCI** | | | | | | | |
| 0 to 1 | 1 | 1 | 1 | 1 | 1 | 1 | 1 |
| 2 to 3 | 1.30 (1.21 to 1.39) | 1.30 (1.05 to 1.61) | 0.92 (0.73 to 1.17) | 1.32 (1.08 to 1.62) | 1.21 (0.99 to 1.48) | 1.26 (1.13 to 1.41) | 1.52 (1.31 to 1.76) |
| 4 to 5 | 1.81 (1.69 to 1.94) | 1.72 (1.39 to 2.12) | 1.37 (1.09 to 1.73) | 1.89 (1.55 to 2.31) | 1.28 (1.04 to 1.58) | 1.74 (1.56 to 1.94) | 2.14 (1.85 to 2.48) |
| ≥6 | 2.51 (2.35 to 2.68) | 2.56 (2.11 to 3.11) | 1.81 (1.47 to 2.24) | 2.40 (1.99 to 2.89) | 2.10 (1.75 to 2.53) | 2.22 (2.01 to 2.46) | 2.63 (2.29 to 3.02) |

[*] Adjusted for sex, age (modeled as restricted cubic splines with 5 knots at ages 30, 49, 64, 73, and 88 years), race, ethnicity, geographical region, urban/rural location, BMI (modeled as restricted cubic splines with 5 knots at BMIs of 21.3, 26.0, 29.0, 32.5, and 39.9 kg/m$^2$), and CCI.

[†] Categorized according to the 10 Federal Regions drawn up by the Federal Emergency Management Agency: 1 (CT, MA, ME, NH, RI, and VT), 2 (NJ, NY, and PR), 3 (DC, DE, MD, PA, VA, and WV), 4 (AL, FL, GA, KY, MS, NC, SC, and TN), 5 (IL, IN, MI, MN, OH, and WI), 6 (AR, LA, NM, OK, and TX), 7 (IA, KS, MO, and NE), 8 (CO, MT, ND, SD, UT, and WY), and 9 (AZ, CA, GU, HI, and NV), and 10 (AK, ID, OR, and WA).

AI/AN, American Indian/Alaska Native; AOR, adjusted odds ratio; BMI, body mass index; CCI, Charlson comorbidity index; PI/NH, Pacific Islander/Native Hawaiian; SARS-CoV-2, Severe Acute Respiratory Syndrome Coronavirus 2; VA, Veterans Affairs.

outside the VA were only captured if documented in the EHR). Therefore, changes in the associations of certain risk factors over time that we observed may also be caused by changes in likelihood of testing for SARS-CoV-2 and identifying positive cases. "Attenuation" over time in risk associated with a high-risk group may also occur if a sufficiently high proportion of that group has already acquired infection and has developed immunity. To avoid this problem, we removed from our cohort persons with known infection prior to the beginning of each monthly/bimonthly observation period. However, persons infected but not tested would still be retained in our cohort and might contribute to attenuation in high-risk groups. Although we adjusted for federal region and urban/rural location, it is possible that some residual confounding by geographic location persists [41]. We did not have data on educational attainment, poverty rates, or income. It is possible that race or ethnicity may be serving as a proxy for low socioeconomic status; the extent to which the associations of race or ethnicity are confounded by socioeconomic status should be examined in future research. Finally, our results apply to US VA enrollees who are predominantly male and have access to comprehensive healthcare. Therefore, they need to be confirmed in other populations. In particular, access to the VA's comprehensive healthcare system has been shown to attenuate some racial and geographic disparities in adverse outcomes in other contexts [42].

In conclusion, strongly positive associations of Black race, AI/AN race, and urban residence with SARS-CoV-2 infection, mortality, and case fatality that were observed early in the pandemic attenuated over time in the VA system. On the other hand, other risk factors, such age, comorbidity burden, Hispanic ethnicity, and obesity, were strongly associated with infection and mortality throughout the observation period. Recognizing the potentially dynamic nature of associations between known risk factors for infection and mortality can help to inform ongoing population-based approaches to prevention and treatment of SARS-CoV-2 as well as providing insights for disparities research in other fields.

## Disclaimers

The contents do not represent the views of the US Department of Veterans Affairs or the US Government.

## Supporting information

**S1 STROBE Checklist. STROBE Statement—Checklist of items that should be included in reports of cohort studies.** STROBE, Strengthening the Reporting of Observational Studies in Epidemiology.
(DOCX)

**S1 Table. AOR\* for interaction term of risk factor and monthly time period of SARS-CoV-2 infection treated as an ordinal variable (risk factor \* time period) among 9.1 million VA enrollees from February 2020 to March 2021, performed as a test of trends over time.** AOR, adjusted odds ratio; SARS-CoV-2, Severe Acute Respiratory Syndrome Coronavirus 2; VA, Veterans Affairs.
(DOCX)

**S2 Table. AOR\* for interaction term of risk factor and monthly time period of SARS-CoV-2–related mortality treated as an ordinal variable (risk factor \* time period) among 9.1 million VA enrollees from February 2020 to February 2021, performed as a test of trends over time.** AOR, adjusted odds ratio; SARS-CoV-2, Severe Acute Respiratory Syndrome Coronavirus 2; VA, Veterans Affairs.
(DOCX)

**S3 Table. Trends over time in the associations (AORs\*) of race with risk of SARS-CoV-2 infection presented separately for persons aged <65 versus ≥65 year among 9.1 million VA enrollees from February 2020 to March 2021.** AOR, adjusted odds ratio;SARS-CoV-2, Severe Acute Respiratory Syndrome Coronavirus 2; VA, Veterans Affairs.
(DOCX)

**S1 Analytic Plan. Changes in risk factors for SARS-CoV-2 infection and mortality over time in a national healthcare system.** SARS-CoV-2, Severe Acute Respiratory Syndrome Coronavirus 2.
(DOCX)

## Author Contributions

**Conceptualization:** George N. Ioannou, Jacqueline M. Ferguson, Ann M. O'Hare, Amy S. B. Bohnert, Lisa I. Backus, Edward J. Boyko, Thomas F. Osborne, C. Barrett Bowling, Denise M. Hynes, Theodore J. Iwashyna, Melody Saysana, Pamela Green, Kristin Berry.

**Data curation:** George N. Ioannou, Pamela Green.

**Formal analysis:** Pamela Green, Kristin Berry.

**Funding acquisition:** George N. Ioannou.

**Investigation:** George N. Ioannou, Ann M. O'Hare.

**Methodology:** George N. Ioannou, Ann M. O'Hare, Kristin Berry.

**Project administration:** Melody Saysana.

**Software:** Pamela Green.

**Writing – original draft:** George N. Ioannou, Kristin Berry.

**Writing – review & editing:** George N. Ioannou, Jacqueline M. Ferguson, Ann M. O'Hare, Amy S. B. Bohnert, Lisa I. Backus, Edward J. Boyko, Thomas F. Osborne, Matthew L.

Maciejewski, C. Barrett Bowling, Denise M. Hynes, Theodore J. Iwashyna, Melody Saysana, Kristin Berry.

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
