## [Editor Report · Decision Letter 0]

16 Jun 2021

Dear Dr Ioannou, 

Thank you for submitting your manuscript entitled "Changes in the associations of race and rurality with SARS-CoV-2 infection and mortality from February 2020 to March 2021" for consideration by PLOS Medicine.

Your manuscript has now been evaluated by the PLOS Medicine editorial staff and I am writing to let you know that we would like to send your submission out for external assessment.

However, we need you to complete your submission by providing the metadata for full assessment. To this end, please login to Editorial Manager where you will find the paper in the 'Submissions Needing Revisions' folder on your homepage. Please click 'Revise Submission' from the Action Links and complete all additional questions in the submission questionnaire.

Please re-submit your manuscript within two working days, i.e. by Jun 18 2021 11:59PM.

Once your full submission is complete, your paper will undergo a series of checks in preparation for assessment. 

Kind regards,

Richard Turner, PhD

rturner@plos.org

---

## [Decision Letter · Decision Letter 1]

22 Jul 2021

Dear Dr. Ioannou,

Thank you very much for submitting your manuscript "Changes in the associations of race and rurality with SARS-CoV-2 infection and mortality from February 2020 to March 2021" (PMEDICINE-D-21-02587R1) for consideration at PLOS Medicine. 

Your paper was discussed among the editors and sent to independent reviewers, including a statistical reviewer. The reviews are appended at the bottom of this email and any accompanying reviewer attachments can be seen via the link below:

[LINK]

In light of these reviews, we will not be able to accept the manuscript for publication in the journal in its current form, but we would like to invite you to submit a revised version that addresses the reviewers' and editors' comments fully. You will appreciate that we cannot make a decision about publication until we have seen the revised manuscript and your response, and we expect to seek re-review by one or more of the reviewers. 

We hope to receive your revised manuscript by Aug 11 2021 11:59PM. Please email us (plosmedicine@plos.org) if you have any questions or concerns.

Please let me know if you have any questions, and we look forward to receiving your revised manuscript. 

Sincerely,

Richard Turner, PhD

rturner@plos.org

To your data statement, please add non-author contact details for those wishing to inquire about access to study data.

Please add a study descriptor to the title following a colon, e.g., "...: a population-based cohort study".

Please quote the setting (e.g., "in the United States") in title and abstract.

Please remove the information on funding and competing interests from the title page. In the event of publication, this information will appear in the article metadata, via entries in the submission form. 

Please combine the "Methods" and "Findings" subsections of your abstract. 

To the new combined subsection, please add a new final sentence, which should begin "Study limitations include ..." or similar and should quote 2-3 of the study's main limitations. 

Please quote aggregate demographic characteristics of study participants in the abstract.

In the abstract and throughout the paper, please quote p values alongside 95% CI, where available.

We suggest beginning the "Conclusions" subsection of the abstract with "In this study, we found that ..." or similar. 

After the abstract, we will need to ask you to add a new and accessible "Author summary" in non-identical prose. You may find it helpful to consult one or two recent research papers in PLOS Medicine to get a sense of the preferred style. 

In the Methods section, please state whether or not the study had a protocol or prespecified analysis plan, and if so attach the document(s) as a supplementary file, referred to in the text.

Please highlight analyses that were not prespecified. 

We suggest substituting "Black race", for example, with "Black ethnicity" throughout.

Throughout the text, please format reference call-outs as follows; "... persons [8,9,14]." (noting the absence of spaces within the square brackets). 

In the reference list, please list no more than 6 author names, followed where appropriate by "et al.".

Noting reference 5 and others, please ensure that references have full access details. 

Please use the journal name abbreviation "PLoS ONE" in the reference list.

Please include a completed checklist for the most appropriate reporting guideline, e.g., STROBE, as a supplementary file, labelled "S1_STROBE_Checklist" or similar and referred to as such in the Methods section. 

In the checklist, please refer to individual items by section (e.g., "Methods") and paragraph number, not by line or page numbers as these will generally change in the event of publication. 

Comments from the reviewers:

*** Reviewer #1: 

I confine my remarks to statistical aspects of this paper. The general approach is fine, but I have one issue to resolve and some comments on the figures.

The authors categorized BMI and age. Categorizing continuous variables is nearly always a bad idea. It increases type I and type II error and imposes arbitrary cutoffs. Here, I see why the authors did it - for ease of presentation of the tables and figures. But I'd like to see models with splines of those variables. You could then make graphs at several ages, to show how things change. So, e..g, the graph on the bottom of p. 18 could show age 18, 30, 40, 50 and 60 (or other ages - the authors could choose). 

For figure 1: First, this is a hidden dual axis graph and these are generally frowned upon. (It's dual axis because the scales are different, per the footnote). What are the authors trying to show? If they want to show each series, they could use two line plots, stacked vertically. If they want to show the ratio of incidence and mortality, they could graph that. Yet another possibility is to have a scatter plot with mortality on one axis, incidence on the other, a dot for each time point, and a line with arrows connecting the dots.

Peter Flom

*** Reviewer #2: 

The analysis is presented in a manner that is clear, concise, and precise. The manuscript provides important findings about changing associations between race and odds of COVID-19 infection and mortality. These findings should be disseminated, after the authors address the following issues concerning their study. This should not require redoing the statistical analysis.

1. The Discussion section should provide greater depth and detail about how the VA population of minority groups are different from the general US population of these groups. Rates of chronic poverty (i.e., persisting over multiple years) and multigenerational poverty are elevated in the US African American population. Chronic, multigenerational poverty may result in health disparities for reasons outlined by Link & Phelan (1995) and their progeny. Is the VA population of African Americans as likely to have experienced this chronic disadvantage as African Americans in general? Are there other reasons that the VA population might consist of a relatively advantaged or privileged group of African Americans (versus the Black population in general), aside from the reason mentioned by the authors (i.e., greater access to healthcare)? The authors might look at whether Blacks in the VA population are more likely to have graduated high school, to currently own a home, or to currently not be in poverty than US Blacks in general.

2. The authors should also address, relatedly, whether racial inequality within the VA population is analogous or parallel to racial inequality in the general population. Though this is not within my expertise, it is my understanding that there is much greater racial diversity among enlisted service members than among officers and especially upper-level officers. To what extent does this inequality in military rank parallel socioeconomic inequality between racial groups in the general population, whereby non-Hispanic Whites are more likely to have greater education and higher status/income jobs? How might differences in racial stratification within the military versus the general population affect differences in susceptibility to COVID-19 morbidity and mortality?

3. The authors do not control for educational attainment or other markers of socioeconomic status (SES) in their analysis. In the general US population, there are major differences between Blacks and Whites in educational attainment, poverty rates, and average income and wealth. In the analysis, race may thus be serving as a proxy for SES, and it may be the case that lower-SES individuals (regardless of race) were more likely to experience COVID morbidity and mortality. The authors could conduct a sensitivity analysis to determine whether results are substantially affected by controlling for an SES indicator, or their lack of controls for SES could be noted prominently as a limitation of the study requiring further exploration in later research.

4. Finally, I found the paper's presentation of its research methods to be somewhat confusing. If I am understanding it correctly, the paper provides results (in the tables) for separate regressions models run for monthly or bimonthly periods. The authors also state that they conducted a regression analysis with data combined from all periods, containing interactions between each predictor variable and a time-period ordinal variable. (This regression with the interaction terms is used—appropriately I believe—to test for change over time in the coefficients of the predictors.) The methods section is not explicit and clear enough in explaining that this single regression model with the interaction terms is distinct from the separate models for the different time periods, or that it is results from these separate regressions that are presented in the graph and tables. Likewise, I was looking for results in the form of a table for the combined regression containing the interaction terms, and I did not find them. This should be added to the paper, in the main text or the appendix. 

*** Reviewer #3: 

This is a useful and important analysis drawing on a richly detailed data source. With that said, I think that the analysis 1. could take more advantage of the available data to give more insight into the drivers of disparity in infection and mortality and how these have changed over time, and 2. requires modification ot the underlying statistical modeling to be suitable for publication. I have two major concerns related to these points:

1. The authors' findings about the differential associations between sociodemographic variables and infections vs. mortality are difficult to interpret, as mortality from SARS-CoV-2 is necessarily contingent on infection. As a result, risk factors for mortality are necessarily correlated with those for infection. Since the authors are analyzing individual-level EHR data, rather than aggregated data, they are able to examine both risks of infection and case-fatality, i.e. the risk of eath on infection. 

Doing this is important as it would allow the specific contributions of factors such as the CCI to mortality to be understood in terms of their specific effects on infection and case-fatality. This would let us see, for example, whether the trend in mortality associated with increasing CCI reflects a stable case-fatality rate across CCI levels, versus some combination of changing infection risks (as evidenced in the relationship between CCI and infection), and case-fatality rates. Analyzing mortality and infection as separate outcomes acts as an unnecessary limitation on the analysis and is one the authors should rectifybefore the manuscript is suitable for publication.

2. The models included in the analysis employ a single parameter for each age group rather than allowing age-specific effects to vary as a function of race/ethnicity. A number of analyses have shown differential age-specific rates of SARS-CoV-2 infection by race/ethnicity, with older African-Americans experiencing much higher rates of infection than same-aged Whites in the early months of the pandemic. Consequently, adjusting for race using a single fixed effect, while not accounting for age x race/ethnicity interactions may result in biased estimates of both the group-specific effects and the associations between age, incidence, and mortality. The authors should provide updated results which include this interaction or show why it is not necessary (e.g. by using posterior predictions from the model which show good coverage of data as a function of age and race/ethnicity)

***

[LINK]

---

## [Decision Letter · Decision Letter 2]

3 Sep 2021

Dear Dr. Ioannou,

Thank you very much for re-submitting your manuscript "Changes in the associations of race and rurality with SARS-CoV-2 infection, mortality and case fatality in the United States from February 2020 to March 2021 : a population-based cohort study" (PMEDICINE-D-21-02587R2) for consideration at PLOS Medicine.

I have discussed the paper with editorial colleagues and it was also seen again by two reviewers. I am pleased to tell you that, provided the remaining editorial and production issues are fully dealt with, we expect to be able to accept the paper for publication in the journal.

[LINK]

Please let me know if you have any questions, and we look forward to receiving the revised manuscript.   

Sincerely,

Richard Turner, PhD

rturner@plos.org

Requests from Editors:

Noting "p<0.01" in your abstract, please quote exact p values or "p<0.001" throughout the paper. 

At the end of the abstract, please remove "even" (from "even reversed ..."). 

The Author summary is quite "data heavy" whereas it is intended to provide an accessible summary of the paper's findings. Therefore, please remove the points quoting quantitative elements also present in the abstract, aiming for three subsections, each of about three points each.

Please also avoid repetition in abstract and author summary.

Please refer to the analysis plan and STROBE checklist by attachment name in your Methods section. 

Please use the journal name abbreviation "PLoS ONE" in the reference list.

Please remove the competing interest information from reference 33, and any other relevant references. 

Comments from Reviewers:

*** Reviewer #2: 

I believe the authors have sufficiently and satisfactorily responded to my prior comments. I recommend that the manuscript is accepted for publication.

*** Reviewer #3: 

Thanks to the authors for their responsiveness to the comments from myself and the other reviewers. I am satisfied with the changes made by the authors and am happy to see this valuable contribution in its current form.

***

[LINK]

---

## [Editor Report · Decision Letter 3]

9 Sep 2021

Dear Dr Ioannou, 

On behalf of my colleagues and the Academic Editor, Dr Zelner, I am pleased to inform you that we have agreed to publish your manuscript "Changes in the associations of race and rurality with SARS-CoV-2 infection, mortality and case fatality in the United States from February 2020 to March 2021: a population-based cohort study" (PMEDICINE-D-21-02587R3) in PLOS Medicine.

Prior to final acceptance, we ask you to remove the author's name from the data statement so as to comply with PLOS' data policy (https://journals.plos.org/plosmedicine/s/data-availability).

PRESS

Sincerely, 

Richard Turner, PhD 

rturner@plos.org